# ENFORCING DELAYED-IMPACT FAIRNESS GUARANTEES

## ABSTRACT

Recent research has shown that seemingly fair machine learning models, when used to inform decisions that have an impact on people's lives or well-being (e.g., applications involving education, employment, and lending), can inadvertently increase social inequality in the long term. Existing fairness-aware algorithms consider static fairness constraints, such as equal opportunity or demographic parity, but enforcing constraints of this type may result in models that have a negative long-term impact on disadvantaged individuals and communities. We introduce `ELF` (Enforcing Long-term Fairness), the first classification algorithm that provides high-confidence fairness guarantees in terms of long-term, or delayed, impact. Importantly, `ELF` solves the open problem of providing such guarantees based only on historical data that includes observations of delayed impact. Prior methods, by contrast, require prior knowledge (or an estimate) of analytical models describing the relationship between a classifier's predictions and their corresponding delayed impact. We prove that `ELF` satisfies delayed-impact fairness constraints with high confidence and that it is guaranteed to identify a fair solution, if one exists, given sufficient data. We show empirically, using real-life data, that `ELF` can successfully mitigate long-term unfairness with high confidence.

## 1 INTRODUCTION

Using machine learning (ML) for high-stakes applications, such as lending, hiring, and criminal sentencing, may potentially harm historically disadvantaged communities (Flage, 2018; Blass, 2019; Bartlett et al., 2021). For example, software meant to guide lending decisions has been shown to exhibit racial bias (Bartlett et al., 2021). Extensive research has been devoted to algorithmic approaches that promote fairness and ameliorate concerns of bias and discrimination for socially impactful applications. Most of this research has focused on the classification setting, in which an ML model must make predictions given information about a person or community.

Most fairness definitions studied in the classification setting are *static*: they do not consider how a classifier's predictions impact the long-term well-being of a community (Liu et al., 2018). In their seminal paper, Liu et al. show that classifiers' predictions that appear fair with respect to static fairness criteria can nevertheless negatively impact the long-term wellness of the community it aims to protect. Importantly, Liu et al., and others investigating long-term fairness (see Section 6), assume that the precise analytical relationship between a classifier's prediction and its long-term impact, or *delayed impact* (DI), is known. *Designing classification algorithms that mitigate negative delayed impact when this relationship is not known, or cannot be computed analytically, has remained an open problem.*

We introduce `ELF` (Enforcing Long-term Fairness), the *first* classification algorithm that solves this open problem. `ELF`, unlike existing methods, does not require access to an analytic model of the delayed impact of a classifier's predictions. Instead, it works under the less strict assumption that the method *only* has access to historical data containing observations of the delayed impact that resulted from predictions of a previously-deployed classifier. Below, we illustrate this setting with an example.

**Loan repayment example.** As a running example, consider a bank that wishes to increase its profit by maximizing successful loan repayments. The bank's decisions are informed by a classifier that predicts repayment success. These decisions may have a delayed impact in terms of the financial well-being of loan applicants, such as their savings rate or debt-to-income ratio, two years after a lending decision is

made. Taking this delayed impact into account is important: when a subset of the population is disadvantaged, the bank may want (or be required by law) to maximize profit subject to a fairness constraint that considers the disadvantaged group's long-term well-being; e.g., a constraint requiring improvement in savings rates two years after a lending decision. Unfortunately, existing methods that address this problem can only be used if analytical models of how repayment predictions affect long-term financial well-being are available. Constructing such models is challenging: many complex factors (e.g., social, economic) influence how different demographic groups may be affected by financial decisions.

ELF, by contrast, can ensure delayed-impact fairness with high confidence as long as the bank can collect data about the delayed impact resulting from predictions made by a previously-deployed classifier. Suppose the bank deployed a classifier, which informed lending decisions, and logged the real-valued savings rate of each client two years later (i.e., logged the observed delayed impact associated with that client). ELF uses this historical data to train a new classifier with high-confidence fairness guarantees in terms of long-term, or delayed, impact.[1] Importantly, ELF works with *all* measures of delayed impact that can be empirically observed or quantified. Appendix A describes other real-life problems where ELF could be applied.

**Contributions.** We present ELF, the first method capable of enforcing DI fairness when the analytical relationship between predictions and DI is not known. To accomplish this, we simultaneously formulate the fair classification problem as both a classification and a reinforcement learning problem—classification for optimizing the primary objective (a measure of classification loss) and reinforcement learning when considering DI. We prove that **1)** the probability that ELF returns a fair model (in terms of DI) is at least $(1-\delta)$, where $\delta$ is a user-specified confidence level; and **2)** given sufficient training data, ELF is able to find and return a solution that is fair if one exists. We empirically analyze ELF's performance on data from the National Data Archive on Child Abuse and Neglect (NDACAN, 2021), while varying both the amount of training data and the influence that a classifier's predictions have on delayed impact.

**Limitations and future work.** ELF requires access to representative historical data—i.e., a dataset with observations of the delayed impact resulting from different predictions. For example, consider a college making admission decisions informed by predictions of students' academic performance. Such predictions may have a delayed impact, e.g., on whether a student will be employed after graduation. The college would only have access to this information about admitted students; in this case, ELF would not be applicable. However, in many important real-life settings *it is* possible to observe the delayed impact of different decisions (see Appendix A). In our lending example, for instance, the bank can observe the delayed impact of lending decisions both for clients who received a loan and those who did not. In Appendix B, we further discuss ELF's limitations.

ELF can be extended in many ways. Section 6 discusses existing methods that address alternative long-term fairness settings; e.g., when multiple prediction steps are involved. Importantly, all of these methods require an analytic expression of the relationship between predictions and delayed impact, while ELF requires access *only* to historical data. As future work, ELF could be extended to tackle these alternative settings.

## 2 PROBLEM STATEMENT

We now formalize the problem of classification with delayed-impact fairness guarantees. As in the standard classification setting, a dataset consists of $n$ data points. Each $i^{\text{th}}$ data point contains $X_i$, a feature vector describing, e.g., a person, and a label $Y_i$. It also contains a set of *sensitive attributes*, such as race and gender. ELF supports an arbitrary number of sensitive attributes, but for brevity our notation uses a single attribute, $T_i$. We assume each data point also contains a prediction, $\widehat{Y}_i^\beta$, made by a stochastic classifier, $\beta$. We call $\beta$ the *behavior model*, defined as $\beta(x, \hat{y}) := \Pr(\widehat{Y}_i^\beta = \hat{y} | X_i = x)$.

Let $I_i^\beta$ be a real-valued *delayed-impact observation* (DIO) resulting from deploying $\beta$ for the person described by the $i^{\text{th}}$ data point. In our running example, $I_i^\beta$ corresponds to the empirically-observed savings rate two years after the prediction $\widehat{Y}_i^\beta$ was used to decide whether the $i^{\text{th}}$ client should get a loan. We assume that larger values of $I_i^\beta$ correspond to better DI. We append $I_i^\beta$ to each data point and

---

[1]In Appendix E we show how ELF can be easily extended to enforce long-term *and* static fairness constraints simultaneously.

thus define the dataset to be a sequence of $n$ independent and identically distributed (i.i.d.) data points: $D := \{(X_i, Y_i, T_i, \widehat{Y}_i^{\beta}, I_i^{\beta})\}_{i=1}^n$. We denote an arbitrary data point in $D$ by suppressing the subscripts.

Our goal is to construct a classification algorithm that takes as input a dataset, $D$, and outputs a new model, $\pi_\theta$, that is as accurate as possible while enforcing delayed-impact constraints.[2] The new model is of the form $\pi_\theta(x, \hat{y}) := \Pr(\widehat{Y}^{\pi_\theta} = \hat{y} | X = x)$, where $\pi_\theta$ is parameterized by a vector $\theta \in \Theta$ (e.g., weights of a neural network), for some feasible set $\Theta$, and where $\widehat{Y}^{\pi_\theta}$ is the prediction made by $\pi_\theta$ given $X$. Like $I_i^{\beta}$, $I_i^{\pi_\theta}$ is the delayed impact observation if the model outputs the prediction $\widehat{Y}_i^{\pi_\theta}$.

We consider the standard classification setting where predictions depend only on the feature vector $X$ (Assumption 1). Furthermore, we assume that the delayed impact of a prediction does not depend on *how* the prediction was made. For example, it makes no difference whether a neural network or a support vector machine made a loan-repayment prediction (Assumption 2).

**Assumption 1.** *A model's prediction $\widehat{Y}^{\pi_\theta}$ is conditionally independent of $Y$ and $T$ given $X$. That is, for all $x, y, t$, and $\hat{y}$, $\Pr(\widehat{Y}^{\pi_\theta} = \hat{y} | X = x, Y = y, T = t) = \Pr(\widehat{Y}^{\pi_\theta} = \hat{y} | X = x)$.*

**Assumption 2.** *For all $x, y, t, \hat{y}$, and $i$,*
$$\Pr(I^{\beta} = i | X = x, Y = y, T = t, \widehat{Y}^{\beta} = \hat{y}) = \Pr(I^{\pi_\theta} = i | X = x, Y = y, T = t, \widehat{Y}^{\pi_\theta} = \hat{y}).$$

Our problem setting can alternatively be described from the reinforcement learning (RL) perspective, where feature vectors are the states of a Markov decision process, predictions are the actions taken by an agent, and DIO is the reward received after the agent takes an action (makes a prediction) given a state (feature vector). From this perspective, Assumption 2 asserts that rewards given an action do not depend on the particular policy that selected the action.

To define long-term fairness, we consider $k$ *delayed-impact objectives*, $g_j : \Theta \to \mathbb{R}, j \in \{1, ..., k\}$, that take as input the parameters, $\theta$, of a classifier, and return a real-valued measurement of fairness in terms of delayed impact. Then, we say that a classifier is fair in the long term iff $g_j(\theta) \leq 0$ for all $j$. To simplify notation, we assume there exists only a single DI objective (i.e., $k = 1$) and later show how to enforce multiple DI objectives (see Algorithm 3). We focus on the case in which each DI objective is based on a conditional expected value having the form

$$g(\theta) := \tau - \mathbf{E}[I^{\pi_\theta} | c(X, Y, T)], \tag{1}$$

where $\tau \in \mathbb{R}$ is a tolerance and $c(X, Y, T)$ is a Boolean conditional relevant to defining the objective. Notice that this form of DI objective allows us to represent DI fairness notions studied in the literature such as Liu et al.'s long-term improvement.[3] To make (1) more concrete, consider our running example, where a bank wishes to enforce a fairness definition that protects a disadvantaged group, $A$. In particular, the bank wishes to ensure that, for a model $\pi_\theta$ being considered, the future financial well-being—in terms of savings rate—of applicants in group $A$ (impacted by $\pi_\theta$'s repayment predictions) does not decline relative to those induced by the previously-deployed model, $\beta$. In this case, $I^{\pi_\theta}$ is the financial well-being of an applicant $t$ months after the loan application, and $c(X, Y, T)$ is the Boolean event indicating if an applicant is in group $A$. Lastly, $\tau$ could represent a threshold on which the bank would like to improve, such as the average financial well-being of type $A$ applicants given historical data collected using $\beta$; i.e., $\tau = \frac{1}{n_-} \sum_{d=1}^n I_d^{\beta} \llbracket T_d = A \rrbracket$, where $\llbracket \cdot \rrbracket$ denotes the Iverson bracket and $n_- = \sum_{d=1}^n \llbracket T_d = A \rrbracket$ is the number of applicants of type $A$. In this setting, the bank is interested in enforcing the following delayed-impact objective: $\mathbf{E}[I^{\pi_\theta} | T = A] \geq \tau$. Then, defining $g(\theta) = \tau - \mathbf{E}[I^{\pi_\theta} | T = A]$ ensures that $g(\theta) \leq 0$ iff the new model $\pi_\theta$ satisfies the DI objective. Notice that an additional constraint of the same form can be added to protect other applicant groups.

## 2.1 Algorithmic properties of interest

As discussed above, to enforce long-term fairness we should ensure that $g(\theta) \leq 0$, since this delayed-impact objective implies that $\theta$ (the model returned by a classification algorithm) is fair with respect to DI. However, this is often not possible, as it requires extensive prior knowledge of how predictions influence DI. Instead, we aim to create an algorithm that uses historical data to reason about its confidence that $g(\theta) \leq 0$. That is, we wish to construct a classification algorithm, $a$, where $a(D) \in \Theta$ is the solution returned by $a$ when given dataset $D$ as input, that satisfies DI constraints of the form

$$\Pr(g(a(D)) \leq 0) \geq 1 - \delta, \tag{2}$$

---

[2]Our algorithm works with arbitrary performance objectives, not just accuracy.

[3]`ELF` can also provide high-confidence guarantees for other forms of DI objectives (Appendix E).

where $\delta \in (0, 1)$ limits the admissible probability that the algorithm returns an unfair model with respect to the DI objective. Algorithms that satisfy (2) are called Seldonian (Thomas et al., 2019).

In practice, there may be constraints that are impossible to enforce simultaneously (Kleinberg et al., 2016), or there may be insufficient data to ensure fairness with high confidence. Then, the algorithm should be allowed to return "No Solution Found" (NSF) instead of a solution it does not trust. Let $\text{NSF} \in \Theta$ and $g(\text{NSF}) = 0$, indicating it is fair for the algorithm to say "I am unable to ensure fairness with the required confidence." Note that a fair algorithm can trivially satisfy (2) by always returning NSF instead of a model. Ideally, if a nontrivial fair solution exists, the algorithm should be able to identify it given enough data. We call this property *consistency* and formally define it in Section 4.

Our goal is to design a fair classification algorithm that satisfies two properties: **1)** the algorithm satisfies (2) and **2)** the algorithm is consistent, i.e., if a nontrivial fair solution exists, the probability that the algorithm returns a fair solution (other than NSF) converges to one as the amount of training data increases. In Section 4, we prove that our algorithm, ELF, satisfies both properties.

## 3    METHODS FOR ENFORCING DELAYED IMPACT

According to our problem statement, a fair algorithm must ensure with high confidence that $g(\theta) \le 0$, where $\theta$ is the returned solution and $g(\theta) = \tau - \mathbf{E}[I^{\pi_\theta} | c(X, Y, T)]$. Because ELF only has access to historical data, $D$, only samples of $I^\beta$ (the delayed impact induced by predictions made by $\beta$) are available. In this section, we show how one can construct i.i.d. estimates of $I^{\pi_\theta}$ using samples collected using $\beta$. Then, we show how confidence intervals can be used to derive high-confidence upper bounds on $g(\theta)$. Lastly, we provide pseudocode for our algorithm, ELF, which satisfies (2).

### 3.1    DERIVING ESTIMATES OF DELAYED IMPACT

First, notice that the distribution of delayed impact observations in $D$ results from predictions made by the model $\beta$. However, our goal is to evaluate the delayed impact of a different model, $\pi_\theta$. This is challenging: given data that includes the DIO resulting from predictions of a previously-deployed model, $\beta$, how to estimate what would the delayed impact be if $\pi_\theta$ were used instead? One naïve solution is to run $\pi_\theta$ on held-out data. However, this would only produce predictions $\widehat{Y}^{\pi_\theta}$, not their corresponding delayed impact; that is, each sample's DIO would still be in terms of $\beta$, not $\pi_\theta$.

We solve this problem using *off-policy evaluation* methods from the RL literature; these use data from running one *policy* (decision-making model) to predict what would happen if a different policy had been used. Specifically, we use *importance sampling* (Precup et al., 2001) to obtain a new random variable, $\hat{I}^{\pi_\theta}$, constructed using data from $\beta$, such that $\hat{I}^{\pi_\theta}$ is an unbiased estimator of $I^{\pi_\theta}$:

$$\mathbf{E}\big[\hat{I}^{\pi_\theta} \,|\, c(X, Y, T)\big] = \mathbf{E}\left[I^{\pi_\theta} \,|\, c(X, Y, T)\right]. \tag{3}$$

In particular, for each data point in $D$, the importance sampling estimator, $\hat{I}^{\pi_\theta}$, weights the observed delayed impacts $I^\beta$ based on how likely the prediction $\widehat{Y}^\beta$ is under $\pi_\theta$. If $\pi_\theta$ would make the label $\widehat{Y}^\beta$ more likely, then $I^\beta$ is given a larger weight (at least one), and if $\pi_\theta$ would make $\widehat{Y}^\beta$ less likely, then $I^{\pi_\theta}$ is given a smaller weight (positive, but less than one). Formally, the importance sampling estimator is $\hat{I}^{\pi_\theta} = \pi_\theta(X, \widehat{Y}^\beta)\beta(X, \widehat{Y}^\beta)^{-1}I^\beta$, where the term $\pi_\theta(X, \widehat{Y}^\beta)/\beta(X, \widehat{Y}^\beta)$ is called the *importance weight*.

Next, we introduce a common assumption in the importance sampling literature: predictions that are possible under $\pi_\theta$ also have non-zero probability of occurring under the behavior model (Assumption 3). In Section 3.3 we discuss a practical way in which this can be ensured. Theorem 1 then establishes that the importance sampling estimator is unbiased, i.e., it satisfies (3).

**Assumption 3** (Support). *For all $x$ and $y$, $\pi_\theta(x, y) > 0$ implies that $\beta(x, y) > 0$.*

**Theorem 1.** *Under assumptions 1–3, $\mathbf{E}[\hat{I}^{\pi_\theta}|c(X, Y, T)] = \mathbf{E}[I^{\pi_\theta}|c(X, Y, T)]$.* **Proof.** *Appendix C.*

In Appendix H, we further discuss the intuition underlying Assumption 3, and others introduced later, and show that they are common in prior work and reasonable in many real-life settings.

## 3.2 Bounds on delayed impact

This section discusses how to use unbiased estimates of $g(\theta)$ together with confidence intervals to derive high-confidence upper bounds on $g(\theta)$. While different confidence intervals for the mean can be used to derive these bounds, to make our method concrete, we consider the specific cases of Student's $t$-test (Student, 1908) and Hoeffding's inequality (Hoeffding, 1963). Consider a vector of $m$ i.i.d. samples $(z_i)_{i=1}^m$ of a random variable $Z$; let the sample mean be $\bar{Z} = \frac{1}{m} \sum_{i=1}^m Z_i$, the sample standard deviation be $\sigma(Z_1, ..., Z_m) = \sqrt{\frac{1}{m-1} \sum_{i=1}^m (Z_i - \bar{Z})^2}$, and $\delta \in (0, 1)$ be a confidence level.

**Property 1.** *If $\sum_{i=1}^m Z_i$ is normally distributed, then*

$$\Pr\left(\mathbf{E}[Z_i] \geq \bar{Z} - \frac{\sigma(Z_1, ..., Z_m)}{\sqrt{m}} t_{1-\delta, m-1}\right) \geq 1 - \delta,$$

*where $t_{1-\delta, m-1}$ is the $1 - \delta$ quantile of the Student's $t$ distribution with $m - 1$ degrees of freedom.*
**Proof.** *See the work of Student (1908).* □

Property 1 can be used to obtain a high-confidence upper bound for the mean of $Z$: $U_{\texttt{ttest}}(Z_1, ..., Z_m) := \bar{Z} + \frac{\sigma(Z_1, ..., Z_m)}{\sqrt{m}} t_{1-\delta, m-1}$. Let $\hat{g}$ be a vector of i.i.d. and unbiased estimates of $g(\theta)$. Once these are computed (using importance sampling; see Section 3.1), they can be provided to $U_{\texttt{ttest}}$ to derive a high-confidence upper bound on $g(\theta)$: $\Pr(\tau - \mathbf{E}[\hat{I}^{\pi_\theta} | c(X, Y, T)] \leq U_{\texttt{ttest}}(\hat{g})) \geq 1 - \delta$. Our strategy for deriving high-confidence upper bounds for delayed-impact objectives is *general* and other confidence intervals for the mean can be used. Student's $t$-test may be used and holds exactly if the distribution of $\sum Z_i$ is normal.[4] Alternatively, in Appendix D we describe a bound based on Hoeffding's inequality (Hoeffding, 1963), which replaces the normality assumption with the weaker assumption that $\hat{g}$ is bounded. This results in a different function for the upper bound, $U_{\texttt{Hoeff}}$.

## 3.3 Complete algorithm

---
**Algorithm 1** ELF
---
**Input**: **1)** $D = \{(X_i, Y_i, T_i, \widehat{Y}_i^\beta, I_i^\beta)\}_{i=1}^n$; **2)** confidence level $\delta$; **3)** tolerance value $\tau$; **4)** behavior model $\beta$; and **5)** $\texttt{Bound} \in \{\texttt{Hoeff}, \texttt{ttest}\}$.
**Output**: Solution $\theta_c$ or NSF.
1: $D_c, D_f \leftarrow \texttt{partition}(D); \quad n_{D_f} = \texttt{length}(D_f); \quad \hat{g} \leftarrow \langle \rangle$
2: $\theta_c \leftarrow \arg\min_{\theta \in \Theta} \texttt{cost}(\theta, D_c, \delta, \tau, \beta, \texttt{Bound}, n_{D_f})$
3: **for** $j \in \{1, ..., n_{D_f}\}$ **do**
4:      Let $(X_j, Y_j, T_j, \widehat{Y}_j^\beta, I_j^\beta)$ be the $j^{\text{th}}$ data point in $D_f$
5:      **if** $c(X_j, Y_j, T_j)$ is True **then** $\hat{g}$.append$\left(\tau - \frac{\pi_{\theta_c}(X_j, \widehat{Y}_j^\beta)}{\beta(X_j, \widehat{Y}_j^\beta)} I_j^\beta\right)$ **end if**
6: **end for**
7: **if** $\texttt{Bound}$ is Hoeff **then** $U = U_{\texttt{Hoeff}}(\hat{g})$ **else if** $\texttt{Bound}$ is ttest **then** $U = U_{\texttt{ttest}}(\hat{g})$ **end if**
8: **if** $U \geq 0$ **then return** NSF **else return** $\theta_c$
---

Algorithm 1 provides pseudocode for ELF. Our algorithm has three main steps. First, the dataset $D$ is divided into two datasets (line 1). In the second step, *candidate selection* (line 2), the first dataset, $D_c$, is used to find and train a model, called the *candidate solution*, $\theta_c$. This step is detailed in Algorithm 2 (see Appendix I). In the *fairness test* (lines 3–8), the dataset $D_f$ is used to compute unbiased estimates of $g(\theta_c)$ using the importance sampling method described in Section 3.1. These estimates are used to calculate a $(1-\delta)$-confidence upper bound, $U$, on $g(\theta_c)$, using Hoeffding's inequality or Student's $t$-test (line 7). Finally, $U$ is used to determine whether $\theta_c$ or NSF is returned.

Recall that for importance sampling to produce unbiased estimates of $g(\theta_c)$, Assumption 3 must hold. To ensure this, we restrict $\Theta$ to only include solutions that satisfy Assumption 3:

**Assumption 4.** *Every $\theta \in \Theta$ satisfies Assumption 3.*

This can be achieved by ensuring that $\beta$ has full support: for all $x$ and $\hat{y}$, $\beta(x, \hat{y}) > 0$. Many supervised learning algorithms already place non-zero probability on every label (e.g., when using Softmax layers in neural networks); in these commonly-occurring cases, Assumption 4 is trivially satisfied.

---
[4]By the central limit theorem, this approximation is reasonable for sufficiently large $m$.

## 4 THEORETICAL RESULTS

This section shows that **1)** `ELF` ensures delayed impact fairness with high confidence; that is, it is guaranteed to satisfy DI constraints as defined in (2); and **2)** given reasonable assumptions about the DI objectives, `ELF` is consistent. To begin, we make an assumption related to the confidence intervals used to bound $g(\theta_c)$, where $\theta_c$ is returned by candidate selection. Specifically, we assume that the requirements related to Student's $t$-test (Property 1) or Hoeffding's inequality (Property 2; see Appendix D) are satisfied. Let $\mathrm{Avg}(Z) = \frac{1}{n_Z} \sum_{i=1}^{n_Z} Z_i$ be the average of a size $n_Z$ vector $Z$.

**Assumption 5.** *If* `Bound` *is* `Hoeff`*, then for all* $j \in \{1, ..., k\}$*, each estimate in* $\hat{g}_j$ *(in Algorithm 3) is bounded in some interval* $[a_j, b_j]$*. If* `Bound` *is* `ttest`*, then each* $\mathrm{Avg}(\hat{g}_j)$ *is normally distributed.*[5]

**Theorem 2.** *Let* $(g_j)_{j=0}^k$ *be a sequence of DI constraints, where* $g_j : \Theta \to \mathbb{R}$*, and let* $(\delta_j)_{j=1}^k$ *be the corresponding confidence levels, where each* $\delta_j \in (0, 1)$*. Under Assumptions 1–5, and if algorithm* $a$ *is Algorithm 3, then for all* $j \in \{1, ..., k\}$*,* $\Pr(g_j(a(D)) \leq 0) \geq 1 - \delta_j$*. **Proof.** See Appendix F.* □

`ELF` satisfies Theorem 2 if the solutions it produces satisfy (2), i.e., if $\forall j \in \{1, ..., k\}$, $\Pr(g_j(a(D)) \leq 0) \geq 1 - \delta_j$, where $a$ is Algorithm 3. Because Algorithm 3 is an extension of Algorithm 1 to multiple constraints, it suffices to show that Theorem 2 holds for Algorithm 3. Next, we show that `ELF` is consistent, i.e., that when a fair solution exists, the probability that the algorithm returns a solution other than `NSF` converges to 1 as the amount of training data goes to infinity:

**Theorem 3** (Consistency guarantee)**.** *If Assumptions 1–8 (6–8 are given in Appendix G) hold, then* $\lim_{n \to \infty} \Pr(a(D) \neq \mathtt{NSF}, g(a(D)) \leq 0) = 1$*. **Proof.** Metevier et al. provide a similar proof for a Seldonian contextual bandit algorithm. Appendix G adapts their proof.*

Proving Theorem 3 requires mild assumptions: **1)** that the cost function used to evaluate classifiers is smooth; **2)** that at least one fair solution exists that is not on the fair-unfair boundary; and **3)** that the sample performance of a classifier converges to its true expected performance given enough data.

## 5 EMPIRICAL EVALUATION

We empirically investigate three research questions: **RQ1:** Does `ELF` enforce DI constraints with high probability, while existing fairness-aware algorithms do not? **RQ2:** What is the cost (e.g., in terms of accuracy) of enforcing DI constraints? **RQ3:** How does `ELF` perform when predictions have little influence on DI relative to other factors outside of the model's control?

We consider a classifier tasked with making predictions about people in the United States foster care system; for example, whether youth currently in foster care are likely to get a job in the near future. These predictions may have a delayed impact on the person's life if, for instance, they influence whether that person receives additional financial aid. Here, the goal is to ensure that a classifier is fair with respect to DI when considering race. Our experiments use two data sources from the

---

[5]`ELF` works with any confidence intervals for the mean. Assumption 5 is reasonable in many real-life settings, and Hoeffding's inequality and Student's $t$-test are effective and commonplace in the sciences (Appendix H).

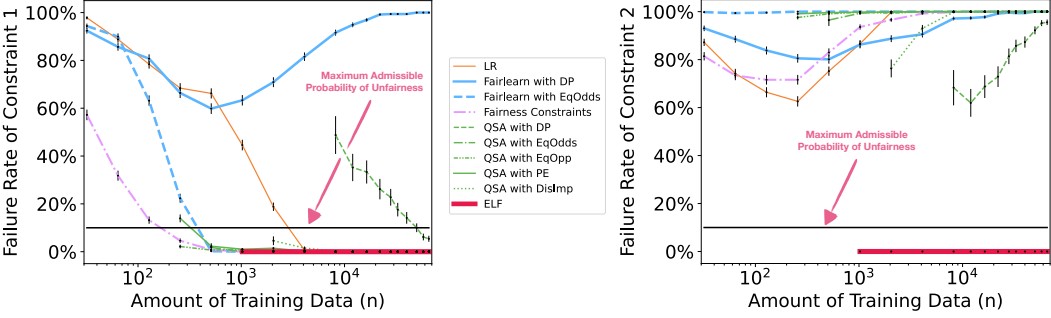

Figure 1: Methods' failure rates w.r.t. DI constraints associated with White people (left) and Black people (right). Black lines show the maximum admissible probability of unfairness, $\delta_0 = \delta_1 = 10\%$.

National Data Archive on Child Abuse and Neglect (NDACAN, 2021): **1)** the Adoption and Foster Care Analysis and Reporting System, containing demographic and foster care-related information about youth; and **2)** the National Youth in Transition Database (Services and Outcomes), containing information about the financial and educational status and well-being of youth over time and during their transition from foster care to independent adulthood.

In this setting, the feature vector $X$ contains five attributes related to the job and educational status of a person in foster care. The sensitive attribute, $T$, corresponds to race—whether a person identifies as White or Black. The classifier is tasked with predicting a binary label, $Y$, denoting whether a person has a full-time job after leaving the program. We modify this dataset by resampling the original labels in a way that ensures the likelihood of being employed after leaving the program depends on *both* $X$ and $T$. The behavior model, $\beta$, corresponds to a logistic regression classifier.

To investigate our research questions, we evaluate `ELF` in settings where predictions made by a classifier may have different levels of influence on DI. We model such settings by constructing a parameterized and synthetic definition of DI. Let $I_i^\psi$ be the delayed impact observation for person $i$ if a classifier $\psi$ outputs the prediction $\widehat{Y}_i^\psi$ given $X_i$. Here, $\psi(x, \hat{y}) := \Pr(\widehat{Y}_i^\psi = \hat{y} | X_i = x)$. We define $I_i^\psi$ as

$$I_i^\psi = \begin{cases} \alpha \widehat{Y}_i^\psi + (1 - \alpha)\mathcal{N}(2, 0.5) & \text{if } T_i = 0 \\ \alpha \widehat{Y}_i^\psi + (1 - \alpha)\mathcal{N}(1, 1) & \text{if } T_i = 1, \end{cases} \tag{4}$$

where $\alpha$ regulates whether DI is strongly affected by a classifier's predictions or if predictions have little to no delayed impact. We refer to the former setting as one with *high prediction-DI dependency* and to the latter setting as one with *low prediction-DI dependency*. As $\alpha$ goes to zero, predictions made by the classifier have no DI. In our experiments, we vary $\alpha$ from zero to one in increments of $0.1$; for each value of $\alpha$, we construct a corresponding dataset by using (4) to assign a DIO to each instance in the foster care dataset. When doing so, $\psi$ is defined as the behavior model, $\beta$.

We wish to guarantee with high probability that the DI caused by a new classifier, $\pi_\theta$, is better than the DI resulting from the currently deployed classifier, $\beta$. This guarantee should hold simultaneously for both races: White (instances where $T = 0$) and Black (instances where $T = 1$). We model this requirement via two DI objectives, $g_0$ and $g_1$. Let $t \in \{0, 1\}$ and $g_t(\theta) := \tau_t - \mathbf{E}[I^{\pi_\theta} | T = t]$, where $\tau_t = \frac{1}{n_t}\sum_{d=1}^n I_d^\beta [\![T_d = t]\!]$ is the average DIO caused by $\beta$ on people of race $T = t$ and where $n_t = \sum_{d=1}^n [\![T_d = t]\!]$. The confidence levels $\delta_0$ and $\delta_1$ associated with these objectives are set to $0.1$.

**RQ1: Preventing delayed-impact unfairness.** To study RQ1, we evaluate whether `ELF` can prevent DI unfairness with high probability, and whether existing algorithms fail. `ELF` is the *first* method capable of ensuring delayed-impact fairness when an analytical model describing the relationship between predictions and DI is not known, and the algorithm *only* has access to historical data. To the best of our knowledge, no other method in the literature ensures DI fairness in this setting (see Section 6). Thus, we compare `ELF` with the closest fairness-aware methods that do not require a model of the prediction-DI dependency nor assume that one can be estimated. In particular, we consider three state-of-the-art fairness-aware algorithms that are designed to enforce static constraints: **1)** Fairlearn (Agarwal et al., 2018), **2)** Fairness Constraints (Zafar et al., 2017), and **3)** quasi-Seldonian algorithms (QSA) (Thomas et al., 2019). We consider five static fairness constraints: demographic parity (DP), equalized odds (EqOdds), disparate impact (DisImp), equal opportunity (EqOpp), and predictive equality (PE) (Chouldechova, 2017; Dwork et al., 2012; Hardt et al., 2016).[6] We also compare `ELF` with a baseline fairness-unaware algorithm: logistic regression (LR).

This section studies how often each fairness-aware algorithm returns an unfair model (with respect to the DI constraints) as a function of the amount of training data, $n$. We refer to the probability that an algorithm returns an unfair model as its *failure rate*. To measure the failure rate, we compute how often the classifiers returned by each algorithm are unfair when evaluated on a significantly larger dataset, to which the algorithms do not have access during training time. To investigate how failure rates are influenced by the level of prediction-DI dependency, we vary $\alpha$ between 0 and 1. Due to space constraints, below we discuss one representative experiment conducted by setting $\alpha = 0.9$. The qualitative behavior of all algorithms for other values of $\alpha$ is similar. The complete set of results and details about our `ELF` implementation can be found in Appendix J.

Figure 1 presents the failure rate of each algorithm as a function of the amount of training data. We computed all failure rates and corresponding standard errors over 500 trials. Notice that the solutions

---

[6]We task each competing method with enforcing the constraints analyzed in those methods' original papers.

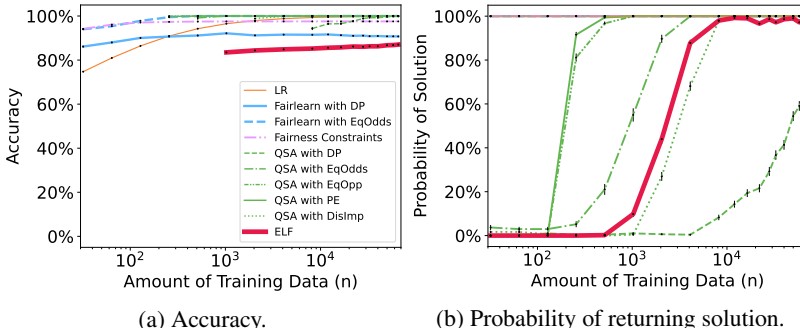

(a) Accuracy.

(b) Probability of returning solution.

Figure 2: On the left, the accuracy of solutions returned by algorithms (subject to different fairness constraints) as a function of $n$. On the right, the probability that these algorithms return a solution.

returned by `ELF` are *always fair* with respect to the DI constraints.[7] This is consistent with `ELF`'s theoretical guarantees, which ensure with high probability that the solutions it returns satisfy all fairness constraints. Existing methods that enforce static fairness criteria, by contrast, either **1)** *always fail* to satisfy both DI constraints; or **2)** *always fail* to satisfy one of the DI constraints—the one related to delayed impact on Black youth—while often failing to satisfy the other constraint.

RQ1: *Our experiment supports the hypothesis that with high probability* `ELF` *is fair with respect to DI objectives, and that comparable fairness-aware techniques do not ensure delayed impact fairness.*

**RQ2: The cost of ensuring delayed-impact fairness.** Previously, we showed that `ELF` is capable of satisfying DI constraints with high probability. Depending on the data domain, this may come at a cost. First, there may be a trade-off between satisfying fairness constraints and optimizing accuracy. In Appendix E we show how `ELF` can be tasked with satisfying DI fairness constraints while also bounding accuracy loss. Here, we investigate the impact that enforcing DI constraints has on accuracy. Figure 2a presents the accuracy of classifiers returned by different algorithms as a function of $n$.[8] In these experiments, we bound accuracy loss via an additional constraint requiring that `ELF`'s solutions have accuracy of at least 75%. Under low-data regimes ($n$=1,000), `ELF`'s accuracy is 83%, while competing methods (with no DI fairness guarantees) have accuracy higher than 90%. Importantly, however, notice that whenever competing methods have higher accuracy than ours, they *consistently return unfair solutions*. `ELF`, by contrast, ensures that *all fairness constraints are satisfied* with high probability and always returns solutions with accuracy above the specified threshold (see Figure 1). Furthermore, notice that as $n$ increases, `ELF`'s accuracy approaches that of the other techniques.

Second, there may be a trade-off between the amount of training data and the confidence that a fair solution has been identified. Recall that some methods (including `ELF`) may not return a solution if they cannot ensure fairness with high confidence. Here we study how often each algorithm identifies a candidate solution as a function of $n$. Figure 2b shows that `ELF` returns solutions with 91% probability when given just $n$=4,096 samples. As $n$ increases, the probability of `ELF` returning solutions increases rapidly. Although three competing techniques (Fairlearn, Fairness Constraints, and LR) always return solutions, independently of the amount of training data, these solutions *never satisfy both DI constraints* (see Figure 1). QSA often returns candidate solutions with less training data than `ELF`; these solutions, however, also *fail to satisfy both DI constraints* simultaneously.

RQ2: *While there is a cost to enforcing DI constraints depending on the data domain,* `ELF` *succeeds in its primary objective: to ensure DI fairness with high probability, without requiring unreasonable amounts of data, while also bounding accuracy loss.*

**RQ3: Varying prediction-DI dependency.** Finally, we investigate `ELF`'s performance (in terms of failure rate, probability of returning solutions, and accuracy) in settings with varied levels of prediction-DI dependency. These include challenging cases where predictions have little influence on DI relative to other factors outside of the model's control.

---

[7] `ELF` does not return solutions with $n < 1,000$ because it cannot ensure DI fairness with high confidence.

[8] As before, we use $\alpha$=0.9 and 500 trials. Results for other values of $\alpha$ are in Appendix J.

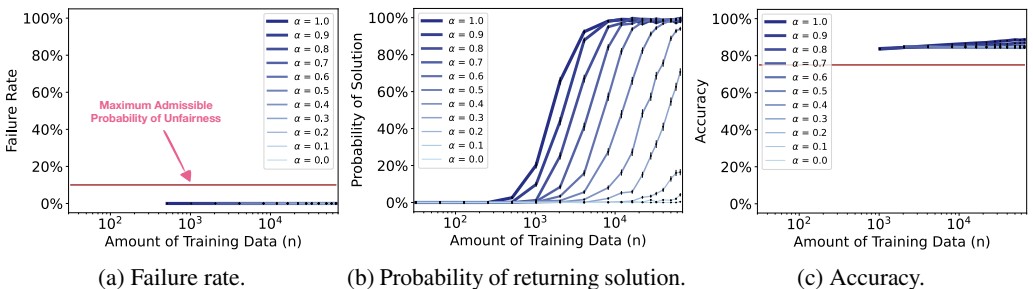

| (a) Failure rate. | (b) Probability of returning solution. | (c) Accuracy. |

Figure 3: `ELF`'s performance in settings with different levels of prediction-DI dependency.

We first study `ELF`'s failure rate (probability that the returned model is unfair) for different values of $\alpha$. Figure 3a shows that `ELF` *never* returns unfair models, independent of $\alpha$, confirming empirically that `ELF`'s high-probability fairness guarantees hold in settings with a wide range of qualitatively different DI characteristics. Next, we investigate how often `ELF` identifies and returns a solution for various values of $\alpha$. We expect that if the predictions made by a classifier have little to no DI (i.e., for low values of $\alpha$), it becomes harder—if not impossible—for `ELF` to be confident that it has identified a model satisfying all DI constraints. This is because it becomes harder to differentiate the small delayed impact of predictions from noise. In these cases, we expect it would be less likely for `ELF` to return solutions. Figure 3b illustrates this behavior. Notice that `ELF` returns solutions under all $\alpha > 0$ if given sufficient training data. However, as expected, the probability that it returns a solution decreases as $\alpha$ approaches zero. Lastly, we investigate how the amount of prediction-DI dependency affects the accuracy of `ELF`'s solutions. Figure 3c shows the model accuracy, for various values of $\alpha$, as a function of $n$. The accuracy trade-off is more evident when `ELF` must satisfy challenging DI objectives, as $\alpha$ approaches zero. In such cases, accuracy decreases from 90% to 84%. Importantly, however, even though a trade-off exists, notice that our method is successful at bounding the accuracy of returned solutions while ensuring DI fairness constraints with high confidence.

RQ3: *Our experiments confirm that `ELF` performs well in a wide range of settings, with various levels of prediction-DI dependency. Even though ensuring fairness may impact accuracy and the probability of finding solutions, these unavoidable, domain-specific trade-offs do not affect `ELF`'s fairness guarantees. In our experiments, all returned models satisfy both DI constraints.*

## 6 RELATED WORK

Most prior work on the social implications of ML study static fairness without considering the long-term impact of model decisions (Dwork et al., 2012; Hardt et al., 2016; Zafar et al., 2017). Recent research, however, has focused on long-term fairness (Hu & Chen, 2018a;b; Liu et al., 2018; D'Amour et al., 2020; Heidari et al., 2019; Zhang et al., 2020a). We build upon this work and present the first method that provides high-confidence delayed-impact fairness guarantees when only historical data is available, and the relationship between predictions and delayed impact is not known.

We now briefly discuss the methods that are most similar to `ELF`. These, and other related techniques, are detailed in Appendix K. Wen et al. (2019) introduce a method that ensures that static fairness constraints (e.g., demographic parity) hold for all time during a sequence of decisions. We study the orthogonal problem of ensuring fairness with respect to user-defined delayed impact measures. The method proposed by Tang et al. (2020) considers the online multi-armed bandit setting; this algorithm does not differentiate individuals within a group while making decisions. We, by contrast, tackle the problem of high-confidence fairness in the classification setting. Zhang et al. (2020b) investigate the long-term effects of repeatedly deploying myopic policies that optimize static fairness constraints. Importantly, they assume knowledge of analytical models characterizing whether an individual will be qualified (e.g., likely to repay a loan) at a given time. Ge et al. (2021) and Hu & Zhang (2022) train classifiers that satisfy static fairness constraints in non-stationary settings (e.g., recommendation systems where a person's interests may change over time). Importantly, however, both Ge et al. (2021) and Hu & Zhang (2022) require knowledge of analytic models of the environment. `ELF`, by contrast, does not require a model or simulator of the environment, nor prior knowledge about the relationship between a classifier's predictions and the resulting delayed impact.

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

## A  OTHER MOTIVATING EXAMPLES FOR ELF

In our paper we discussed two important real-life examples where our method could be applied. First, we introduced a motivating problem where a bank wishes to take into account the delayed impact of lending decisions on a disadvantaged group. Secondly, in our empirical evaluation we considered the delayed impact of providing financial aid to youth in foster care. Here, we discuss three additional examples of possible applications of ELF in real-life settings:

- Consider a university that has a 1-on-1 tutoring program. However, the university has limited resources and cannot offer tutoring for all students. To select which students should participate in the program, the university's decision is based on GPA predictions. Receiving tutoring (or not) can influence the chances of a student graduating from college. In this case, the delayed impact observation could be whether a student graduated or not. The university has access to this DIO both for students that received tutoring and for those who did not.

- Consider a police department deciding which crime prevention strategy to use in each district of a city, based on predictions about crime recidivism. DI could be the average incarceration rate in each district two years after this decision; it could indicate, e.g., that people of a particular race are unfairly affected by the strategies deployed by the policy department.

- Assume that medical decisions are influenced by predictions of whether a person qualifies for high-risk care management. These predictions may have a DI on a person's health; e.g., the severity of chronic illnesses two years in the future. Different demographic groups may be affected differently in the long term.

## B  LIMITATIONS

ELF requires access to representative historical data—i.e., a dataset with observations of the delayed impact resulting from different predictions. For example, consider a college whose admissions decisions are informed by a classifier predicting students' academic performance. These predictions may have a delayed impact, e.g., on whether the student will be employed after graduation. However, the college would only have access to this information about admitted students; in this case, ELF would not be applicable. Nonetheless, in many important real-life settings *it is* possible to observe the delayed impact of different decisions (see Appendix A). For instance, in the lending example described in the paper, the bank can observe the delayed impact of lending decisions both for clients who received a loan and those who did not.[9]

Furthermore, ELF's high probability fairness guarantees only hold if the world has not changed between the time training data was collected and the time the trained classifier is deployed. While this (stationarity) assumption is common in ML, it may be challenging in our setting since gathering data that includes measures of long-term impact requires that a correspondingly long duration of time

---

[9]Even though ELF may be affected by this limitation, we argue that the assumption that our algorithm has access to a representative dataset is significantly less strict than the requirements and assumptions made by all existing techniques. Current state-of-the-art methods require access to complete and accurate models of the environment. Ensuring access to complete models, in practical applications, is qualitatively harder than ensuring access to representative datasets. Constructing models of this type could involve, e.g., designing analytic expressions describing how economic and social factors (both observed and latent) might affect a given person's financial well-being in the future.

has passed. While providing guarantees when nonstationarity occurs is important future work, in this paper we focus on the important first step of providing the first classification algorithm that provides delayed-impact fairness guarantees in the stationary setting.

## C   PROOF OF THEOREM 1

*Proof.* At a high level, we start with $\mathbf{E}[\hat{I}^{\pi_\theta}|c(X,Y,T)]$ and, through a series of transformations involving substitution, laws of probability, and Assumptions 1–3, obtain $\mathbf{E}[I^{\pi_\theta}|c(X,Y,T)]$. To simplify notation, throughout this proof we let $C = c(X,Y,T)$. Also, for any random variable $Z$, let $\mathrm{supp}(Z)$ denote the support of $Z$ (e.g., if $Z$ is discrete, then $\mathrm{supp}(Z) = \{z : \Pr(Z=z) > 0\}$). To begin, we substitute the definition of $\hat{I}^{\pi_\theta}$ and expand this expression using the definition of expected value:

$$\mathbf{E}[\hat{I}^{\pi_\theta}|C] = \mathbf{E}\left[\frac{\pi_\theta(X,\widehat{Y}^\beta)}{\beta(X,\widehat{Y}^\beta)}I^\beta \,\middle|\, C\right]$$

$$= \sum_{(x,y,t,\hat{y},i)\in\mathrm{supp}(X,Y,T,\widehat{Y}^\beta,I^\beta)} \Pr(X{=}x, Y{=}y, T{=}t, \widehat{Y}^\beta{=}\hat{y}, I^\beta{=}i|C)\frac{\pi_\theta(x,\hat{y})}{\beta(x,\hat{y})}i. \tag{5}$$

Using the chain rule repeatedly, we can rewrite the joint probability in (5) as follows:

$$\Pr(X{=}x, Y{=}y, T{=}t, \widehat{Y}^\beta{=}\hat{y}, I^\beta{=}i|C)$$
$$= \Pr(I^\beta{=}i|X{=}x, Y{=}y, T{=}t, \widehat{Y}^\beta{=}\hat{y}, C)\Pr(X{=}x, Y{=}y, T{=}t, \widehat{Y}^\beta{=}\hat{y}|C)$$
$$= \Pr(I^\beta{=}i|X{=}x, Y{=}y, T{=}t, \widehat{Y}^\beta{=}\hat{y}, C)\Pr(\widehat{Y}^\beta{=}\hat{y}|X{=}x, Y{=}y, T{=}t, C)\Pr(X{=}x, Y{=}y, T{=}t|C).$$

Under Assumption 1, $\Pr(\widehat{Y}^\beta{=}\hat{y}|X{=}x, T{=}t, Y{=}y, C) = \Pr(\widehat{Y}^\beta{=}\hat{y}|X{=}x)$, which is the definition of $\beta(x,\hat{y})$. We perform this substitution and simplify by cancelling out the $\beta$ terms:

$$\mathbf{E}[\hat{I}^{\pi_\theta}|C] = \sum_{(x,y,t,\hat{y},i)\in\mathrm{supp}(X,Y,T,\widehat{Y}^\beta,I^\beta)} \Pr\left(I^\beta{=}i|X{=}x, Y{=}y, T{=}t, \widehat{Y}^\beta{=}\hat{y}, C\right)\beta(x,\hat{y})\Pr\left(X{=}x, Y{=}y, T{=}t|C\right)\frac{\pi_\theta(x,\hat{y})}{\beta(x,\hat{y})}i$$

$$= \sum_{(x,y,t,\hat{y},i)\in\mathrm{supp}(X,Y,T,\widehat{Y}^\beta,I^\beta)} \Pr\left(I^\beta{=}i|X{=}x, Y{=}y, T{=}t, \widehat{Y}^\beta{=}\hat{y}, C\right)\Pr\left(X{=}x, Y{=}y, T{=}t|C\right)\pi_\theta(x,\hat{y})i. \tag{6}$$

Note that under Assumption 1, $\pi_\theta(x,\hat{y})$ can be rewritten as $\Pr(\widehat{Y}^{\pi_\theta}{=}\hat{y}|X{=}x, Y{=}y, T{=}t, C)$. Using the multiplication rule of probability, we can combine this term with the $\Pr(X{=}x, Y{=}y, T{=}t|C)$ term in (6) to obtain the joint probability $\Pr(X{=}x, Y{=}y, T{=}t, \widehat{Y}^{\pi_\theta}{=}\hat{y}|C)$. By Assumption 2, we can substitute $\Pr(I^\beta{=}i|X{=}x, Y{=}y, T{=}t, \widehat{Y}^\beta{=}\hat{y}, C)$ for $\Pr(I^{\pi_\theta}{=}i|X{=}x, Y{=}y, T{=}t, \widehat{Y}^{\pi_\theta}{=}\hat{y}, C)$. We substitute these terms into (6) and apply the multiplication rule of probability once more:

$$\mathbf{E}[\hat{I}^{\pi_\theta}|C] = \sum_{(x,y,t,\hat{y},i)\in\mathrm{supp}(X,Y,T,\widehat{Y}^\beta,I^\beta)} \Pr(I^{\pi_\theta}{=}i|X{=}x, Y{=}y, T{=}t, \widehat{Y}^{\pi_\theta}{=}\hat{y}, C)\Pr(X{=}x, Y{=}y, T{=}t, \widehat{Y}^{\pi_\theta} = \hat{y}|C)i$$

$$= \sum_{(x,y,t,\hat{y},i)\in\mathrm{supp}(X,Y,T,\widehat{Y}^\beta,I^\beta)} \Pr(X{=}x, Y{=}y, T{=}t, \widehat{Y}^\beta{=}\hat{y}, \hat{I}^{\pi_\theta}{=}i|C)i. \tag{7}$$

Finally, notice that by Assumption 3, $\mathrm{supp}(\widehat{Y}^{\pi_\theta}) \subseteq \mathrm{supp}(\widehat{Y}^\beta)$, and so $\mathrm{supp}(I^{\pi_\theta}) \subseteq \mathrm{supp}(I^\beta)$. So, we can rewrite (7) as

$$\sum_{(x,y,t,\hat{y},i)\in\mathrm{supp}(X,Y,T,\widehat{Y}^{\pi_\theta},I^{\pi_\theta})} \Pr(X{=}x, Y{=}y, T{=}t, \widehat{Y}^{\pi_\theta}{=}\hat{y}, \hat{I}^{\pi_\theta}{=}i|C)i.$$

By the definition of expectation, this is equivalent to $\mathbf{E}\left[I^{\pi_\theta}|C\right]$. Therefore, we have shown that $\mathbf{E}[\hat{I}^{\pi_\theta}|C] = \mathbf{E}[I^{\pi_\theta}|C]$. □

## D  BOUNDS ON DELAYED IMPACT USING HOEFFDING'S INEQUALITY

This section focuses on how one can use the unbiased estimates of $g(\theta)$ together with Hoeffding's inequality (Hoeffding, 1963) to derive high-confidence upper bounds on $g(\theta)$. Given a vector of $m$ i.i.d. samples $(Z_i)_{i=1}^m$ of a random variable $Z$, let $\bar{Z} = \frac{1}{m}\sum_{i=1}^m Z_i$ be the sample mean, and let $\delta \in (0, 1)$ be a confidence level.

**Property 2** (Hoeffding's Inequality). *If* $\Pr(Z \in [a, b]) = 1$, *then*

$$\Pr\left(\mathbf{E}[Z_i] \geq \bar{Z} - (b - a)\sqrt{\frac{\ln(1/\delta)}{2m}}\right) \geq 1 - \delta.$$

**Proof.** *See the work of Hoeffding (1963).*  □

Property 2 can be used to obtain a high-confidence upper bound for the mean of $Z$:

$$U_{\texttt{Hoeff}}(Z_1, Z_2, ..., Z_m) := \bar{Z} + (b-a)\sqrt{\frac{\log(1/\delta)}{(2m)}}.$$

Let $\hat{g}$ be a vector of i.i.d. and unbiased estimates of $g(\theta)$. Once these are procured (using importance sampling as described in Section 3.1), they can be provided to $U_{\texttt{Hoeff}}$ to derive a high-confidence upper bound on $g(\theta)$:

$$\Pr\left(\tau - \mathbf{E}\left[\hat{I}^{\pi_\theta}\Big| c(X, Y, T)\right] \leq U_{\texttt{Hoeff}}(\hat{g})\right) \geq 1 - \delta.$$

Notice that using Hoeffding's inequality to obtain the upper bound requires the assumption that $\hat{g}$ is bounded.

## E  EXTENSIONS OF ELF

In this section we discuss how ELF can be extended to provide similar high-confidence guarantees for the regression setting, for the classification setting with static fairness constraints, and for definitions of delayed impact objectives beyond the form assumed in (1).

### E.1  DELAYED-IMPACT FAIRNESS GUARANTEES IN THE REGRESSION SETTING

In our problem setting, we study fairness with respect to delayed impact in the classification setting, in which the labels $Y$ produced by a model are discrete. However, our method can also be applied in the regression setting, where a (stochastic) regression model produces *continuous* predictions $Y$, instead of discrete labels. To use ELF in this setting, one may adapt Algorithm 2 so that it uses a loss function suitable for regression; e.g., sample mean squared error. Furthermore, notice that the importance sampling technique described in Section 3.1 is still applicable in the regression setting, requiring only minor changes so that it can be used in such a continuous setting. In particular, the importance sampling technique we described can be adapted by replacing summations with integrals, probability mass functions with probability density functions, and probabilities with probability densities. By doing so, all results presented in our work (e.g., regarding the unbiasedness of the importance sampling estimator) carry to the continuous case. Notice, finally, that in order to apply ELF in the regression setting, the behavior model, $\beta$, and the new candidate model, $\pi_\theta$, must be *stochastic* regression models—this is similar to the assumption we made when addressing the classification setting (see the discussion in Section 2).

### E.2  ENFORCING STATIC FAIRNESS CONSTRAINTS OR CONSTRAINTS ON MODEL PERFORMANCE

In Section 3, we showed how users can construct and enforce delayed-impact fairness constraints. However, users might also be interested in simultaneously enforcing additional behavior—for instance, enforcing *static* fairness constraints or constraints on the primary objective (i.e., on the classification or regression performance). Assume, for example, that the bank from our running example has

constructed a delayed-impact fairness constraint of the form in (1), and that it is *also* interested in specifying an additional constraint that lower-bounds the model's performance in terms of accuracy. This could be represented by the following objective:

$$g_{\text{PERF}}(\theta) = \tau_{\text{PERF}} - \mathbf{E}[\text{ACC}_\theta],$$

where $\tau_{\text{PERF}} \in (0, 1)$ represents the minimum required accuracy, and $\text{ACC}_\theta$ is a random variable representing the accuracy of the model $\theta$, i.e., the fraction of predictions that are correct given a random dataset $D$.

Continuing the example, assume that the bank would also like to enforce a *static* fairness constraint: *false positive error rate balance* (FPRB) (Chouldechova, 2017), also known as *predictive equality* (Corbett-Davies et al., 2017). Recall that a classifier satisfies an approximate version of FPRB if the absolute difference between the false positive rates of two demographic groups of loan applicants, A and B, is below some user-specified threshold. The bank could specify this static fairness objective as:

$$g_{\text{FPRB}} = \left| \mathbf{E}\left[\widehat{Y}^{\pi_\theta} = 1 \middle| Y = 0, T = A\right] - \mathbf{E}\left[\widehat{Y}^{\pi_\theta} = 1 \middle| Y = 0, T = B\right] \right| - \epsilon_{\text{FPRB}},$$

where $\epsilon_{\text{FPRB}} \in (0, 1)$ is the threshold of interest.

Notice that—unlike delayed-impact objectives—$g_{\text{PERF}}$ and $g_{\text{FPRB}}$ can be directly computed using only labels $Y$, predictions $\widehat{Y}^{\pi_\theta}$, and the sensitive attribute $T$; that is, information already available in the dataset or directly obtained from the model. In other words, the importance sampling method introduced in Section 3 would not be needed to obtain high-confidence bounds for these metrics. Metevier et al. (2019) and Thomas et al. (2019) present methods to compute high-confidence upper bounds on static fairness constraints and constraints on performance. Notice that $g_{\text{PERF}}$ and $g_{\text{FPRB}}$ are just examples of this type of constraints; the techniques introduced by Metevier et al. (2019) and Thomas et al. (2019) are applicable to more general objectives and constraints. We refer the reader to their work for more details.

To conclude our discussion of this example, notice that—once computed—high-confidence upper bounds on $g_{\text{PERF}}$ and $g_{\text{FPRB}}$ may be used to determine whether a candidate solution should be returned. Similar to line 11 of Algorithm 3, if all computed upper bounds (with respect to the accuracy objective, the predictive equality objective, and the delayed-impact objectives) are less than or equal to zero, then the candidate solution should be returned. Otherwise, `NSF` should be returned.

### E.3  ALTERNATIVE DEFINITIONS OF DELAYED-IMPACT OBJECTIVES

Until now, we have assumed that the delayed-impact objective takes the form of (1). Below, we discuss how users of `ELF` may construct other definitions of delayed-impact objectives, and how our formulation of delayed-impact objective (shown in (1)) is related to the definitions introduced in the work of Liu et al. (2018).

**Connections to the work of Liu et al. (2018).**  Our DI objective (1) has the form $g(\theta) \coloneqq \tau - \mathbf{E}\left[I^{\pi_\theta} | c(X, Y, T)\right]$. This is similar to *long-term improvement*, one of the notions of delayed impact introduced by Liu et al. (2018). Specifically, Liu et al. (2018) define long-term improvement as $\Delta\mu_j > 0$, where for group $j$, $\Delta\mu_j$ is the difference between the DI induced by a previously-deployed model and a new model. In their work, Liu et al. (2018) consider DI to be credit score. To enforce this type of long-term improvement in our framework, we can set $\tau$ to be group $j$'s average credit score under the current model (i.e., under the behavior model, $\beta$) and $\mathbf{E}\left[I^{\pi_\theta} | T = j\right]$ to be the expected credit score of group $j$ under the new model, $\pi_\theta$. Then, $\Delta\mu_j = \mathbf{E}\left[I^{\pi_\theta} | T = j\right] - \tau$. In our framework, a model is fair if $g(\theta) \leq 0$. Setting $g(\theta) = \tau - \mathbf{E}\left[I^{\pi_\theta} | T = j\right]$ implies that the model is fair (i.e., $g(\theta) \leq 0$) if and only if $\theta$ leads to long-term improvement (i.e., iff $\Delta\mu_j > 0$).

We can also use `ELF` to enforce constraints similar to the remaining definitions of delayed-impact objective introduced by Liu et al. (2018); e.g., long-term decline ($\Delta\mu_j < 0$) and long-term stagnation ($\Delta\mu_j = 0$). Notice that long-term decline has a form similar to long-term improvement: $\Delta\mu_j < 0 \implies g(\theta) = \mathbf{E}\left[I^{\pi_\theta} | T = j\right] - \tau$. Alternatively, to enforce long-term stagnation users can set $g(\theta) = \left| \tau - E\left[I^{\pi_\theta} | T = j\right] \right|$. Moreover, to enforce *approximate* long-term stagnation, i.e., $|\Delta\mu| < \epsilon$, for some non-negative threshold $\epsilon$, users may set $g(\theta) = \left| \tau - \mathbf{E}\left[I^{\pi_\theta} | T = j\right] \right| - \epsilon$.

Finally, notice that the definitions of long-term decline and stagnation do not have the same form as (1); nonetheless, these definitions *can* be enforced using the methods introduced in our work—we discuss how to achieve this in the next section.

**Enforcing general definitions of delayed-impact objectives**  To enforce constraints beyond (1), one can combine the importance sampling technique introduced in Section 3 with techniques presented in the work of Metevier et al. (2019). Assume, for example, that the bank in our running example is interested in enforcing that the resulting expected delayed impact of a classifier's predictions is approximately equal for loan applicants of group $A$ and group $B$. This can be represented by the DI objective $g_{\mathrm{DI}}(\theta) = \big| \mathbf{E}\left[I^{\pi_\theta}|T = A\right] - \mathbf{E}\left[I^{\pi_\theta}|T = B\right] \big| - \epsilon$. To enforce the DI constraint considered in this paper (i.e., the one shown in (2)) on the more general types of DI objectives discussed in this appendix (such as $g_{\mathrm{DI}}$), one may combine the techniques we introduced in this paper and the bound-propagation methods introduced by Metevier et al. (2019). At a high-level, `ELF` would, in this case, first compute (as before) unbiased estimates of $\mathbf{E}\left[I^{\pi_\theta}|T = A\right]$ and $\mathbf{E}\left[I^{\pi_\theta}|T = B\right]$ using the importance sampling technique described in Section 3.1. Then, it would use the bound-propagation methods introduced by Metevier et al. (2019) to obtain high-confidence upper bounds on $g_{\mathrm{DI}}(\theta)$. Notice that the discussion above corresponds to just one example of how to deal with alternative delayed-impact objective definitions; in this particular example, $g_{\mathrm{DI}}$. The same general idea and techniques can, however, also be used to deal with alternative definitions of DI objectives that users of `ELF` may be interested in.[10] All other parts of the algorithm would remain the same—e.g., the algorithm would still split the dataset into two, identify a candidate solution, and check whether it passes the fairness test.

**Beyond conditional expectation.**  In Section 2, we assume that $g$ is defined in terms of the conditional expected value of the delayed-impact measure. However, other forms of fairness metrics might be more appropriate for different applications. For example, conditional value at risk (Keramati et al., 2020) might be appropriate for risk-sensitive applications, and the median might be relevant for applications with noisy data (Altschuler et al., 2019). Chandak et al. (2021) introduce off-policy evaluation methods that produce estimates and high-confidence bounds for different distributional parameters of interest, including value at risk, conditional value at risk, variance, median, and interquantile range. These techniques can also be combined with ours to obtain high-confidence upper bounds for metrics other than the conditional expected value of $I^{\pi_\theta}$.

## F  PROOF OF THEOREM 2: FAIRNESS GUARANTEE

This section proves Theorem 2, which is restated below, along with the relevant assumptions.

> **Assumption 1:** A model's prediction $\widehat{Y}^{\pi_\theta}$ is conditionally independent of $Y$ and $T$ given $X$. That is, for all $x, t, y$, and $\hat{y}$,
>
> $$\Pr(\widehat{Y}^{\pi_\theta}=\hat{y}|X=x, Y=y, T=t) = \Pr(\widehat{Y}^{\pi_\theta}=\hat{y}|X=x).$$
>
> **Assumption 2:** For all $x, y, t, \hat{y}, i$,
>
> $$\Pr(I^{\beta}=i|X=x, Y=y, T=t, \widehat{Y}^{\beta}=\hat{y}) = \Pr(I^{\pi_\theta}=i|X=x, Y=y, T=t, \widehat{Y}^{\pi_\theta}=\hat{y}).$$
>
> **Assumption 3 (Support):** For all $x$ and $y$, $\pi_\theta(x,y) > 0$ implies that $\beta(x,y) > 0$.
> **Assumption 4:** Every $\theta \in \Theta$ satisfies Assumption 3.
> **Assumption 5:** If `Bound` is `Hoeff`, then for all $j \in \{1, ..., k\}$, each estimate in $\hat{g}_j$ is bounded in some interval $[a_j, b_j]$. If `Bound` is `ttest`, then each $\mathrm{Avg}(\hat{g}_j)$ is normally distributed.

---

[10]This statement holds assuming that the DI objective of interest satisfies the requirements for the bound-propagation technique to be applicable; for example, that the DI objective can be expressed using elementary arithmetic operations (e.g., addition and subtraction) over *base variables* for which we know unbiased estimators (Metevier et al., 2019). In the case of the DI objectives discussed in this paper, for instance, we can obtain unbiased estimates of the relevant quantities using importance sampling, as discussed in Section 3.1.

> **Theorem 2:** Let $(g_j)_{j=1}^k$ be a sequence of DI constraints, where $g_j : \Theta \to \mathbb{R}$, and let $(\delta_j)_{j=1}^k$ be a corresponding sequence of confidence levels, where each $\delta_j \in (0, 1)$. If Assumptions 1, 2, 4, and 5 hold, then for all $j \in \{1, ..., k\}$,
>
> $$\Pr(g_j(a(D)) \le 0) \ge 1 - \delta_j.$$

We first provide three lemmas that will be used when proving Theorem 2.

**Lemma 1.** *Let $\hat{g}_j$ be the estimates of $g$ constructed in Algorithm 3, and let $D_{f_c}$ be a subdataset of $D_f$ such that a data point $(X, Y, T, \widehat{Y}^\beta, I^\beta)$ is only in $D_{f_c}$ if $c(X, Y, T)$ is true. Then, for all $\theta \in \Theta$, the elements in $\hat{g}_j$ are i.i.d. samples from the conditional distribution of $\hat{g}_j$ given $c(X, Y, T)$.*

*Proof.* To obtain $\hat{g}_j$, each data point in $D_{f_c}$ is transformed into an estimate of $g(\theta)$ using the importance sampling estimate $\tau - \frac{\pi_\theta(X, \widehat{Y}^\beta)}{\beta(X, \widehat{Y}^\beta)} I^\beta$ (Algorithm 3, lines 5–8). Since each element of $\hat{g}_j$ is computed from a single data point in $D_{f_c}$, and the points in $D_{f_c}$ are conditionally independent given $c(X, Y, T)$, it follows that each element of $\hat{g}_j$ is conditionally independent given $c(X, Y, T)$. So, each element of $\hat{g}_j$ can be viewed as an i.i.d. sample from the conditional distribution of $\hat{g}_j$ given $c(X, Y, T)$. $\square$

**Lemma 2.** *Let $\hat{g}_j$ be the estimates of $g$ constructed in Algorithm 3. If Assumptions 1, 2, and 4 hold, then for all $\theta \in \Theta$, each element in $\hat{g}_j$ is an unbiased estimate of $g_j(\theta)$.*

*Proof.* We begin by considering the expected value of any element in $\hat{g}_j$:

$$
\begin{aligned}
\mathbf{E}\left[\tau - \frac{\pi_\theta(X, \widehat{Y}^\beta)}{\beta(X, \widehat{Y}^\beta)} I^\beta \,\middle|\, c(X, Y, T)\right] &= \tau - \mathbf{E}\left[\frac{\pi_\theta(X, \widehat{Y}^\beta)}{\beta(X, \widehat{Y}^\beta)} I^\beta \,\middle|\, c(X, Y, T)\right] \\
&= \tau - \mathbf{E}\left[\hat{I}^{\pi_\theta} \,\middle|\, c(X, Y, T)\right] \\
&= \tau - \mathbf{E}\left[I^{\pi_\theta} \,\middle|\, c(X, Y, T)\right] \qquad (8)\\
&= g_j(\theta).
\end{aligned}
$$

Expression (8) follows from Theorem 1, which relies on Assumptions 1, 2, and 4. Therefore, for all $\theta \in \Theta$, the elements of $\hat{g}_j$ are unbiased estimates of $g_j(\theta)$. $\square$

Let $\theta_c$ be the model returned by candidate selection in Algorithm 3 (line 2), and let $U_j$ be the value of $U$ at iteration $j$ of the for loop (lines 4–10).

**Lemma 3.** *If Assumptions 1, 2, 4, and 5 hold, then the upper bounds $U_j$ calculated in Algorithm 3 satisfy $\forall j \in \{1, ..., k\}$, $\Pr(g_j(\theta_c) > U_j) \le \delta_j$.*

*Proof.* We begin by noting that by Lemma 1, the data points used to construct each $(1-\delta_j)$-probability bound, i.e., the data points in each $\hat{g}_j$, are (conditionally) i.i.d. Because $\theta_c \in \Theta$, by Lemma 2 (which uses Assumptions 1, 2, and 4), we know that each element in $\hat{g}_j$ is an unbiased estimate of $g_j(\theta_c)$. Therefore, Hoeffding's inequality or Student's $t$-test can be applied to random variables that are (conditionally) i.i.d.[11] and unbiased estimates of $g_j(\theta_c)$. Moreover, under Assumption 5, when `Bound` is `Hoeff`, the requirements of Hoeffding's inequality are satisfied (Property 2), and when `Bound` is `ttest`, the requirements of Student's $t$-test are satisfied (Property 1). Therefore, the upper bounds calculated in Algorithm 3 satisfy $\Pr(g_j(\theta_c) > U_j) \le \delta_j$. $\square$

**Proof of Theorem 2**

---

[11] Samples that are conditionally i.i.d. given some event $E$ can be viewed as i.i.d. samples from the conditional distribution. Applying the confidence intervals to these samples therefore provides high-confidence bounds on the *conditional* expected value given the event $E$, which is precisely what we aim to bound.

*Proof.* To show Theorem 2, we prove the contrapositive, i.e., $\forall j \in \{1, ..., k\}, \Pr(g_j(a(D)) > 0) \leq \delta_j$.

Consider the event $\forall j \in \{1, ..., k\}, g_j(a(D)) > 0$. When this event occurs, it is always the case that $a(D) \neq \texttt{NSF}$ (by definition, $g(\texttt{NSF}) = 0$). That is, a nontrivial solution was returned by the algorithm, and for all $j$, $U_j \leq 0$ (line 11 of Algorithm 3). Therefore, (9) (shown below) holds.

$$\Pr(g_j(a(D) > 0) = \Pr(g_j(a(D)) > 0, U_j \leq 0) \tag{9}$$
$$\leq \Pr(g_j(a(D)) > U_j) \tag{10}$$
$$= \Pr(g_j(\theta_c) > U_j) \tag{11}$$
$$\leq \delta_j. \tag{12}$$

Expression (10) is a result of the fact that the joint event in (9) implies the event $(g_j(a(D)) > U_j)$. We substitute $\theta_c$ for $a(D)$ in (11) because the event $\forall j \in \{1, ..., k\}, g_j(a(D)) > 0$ implies that a nontrivial solution, or a solution that is not $\texttt{NSF}$, was returned: $a(D) = \theta_c$. Lastly, (12) follows from Lemma 3. This implies that $\Pr(g_j(a(D) > 0) \leq \delta_j \ \forall j \in \{1, ..., k\}$, completing the proof.

$\square$

## G  PROOF OF THEOREM 3

This section proves Theorem 3, restated below. Metevier et al. (2019) provide a similar proof for a Seldonian contextual bandit algorithm, which we adapt to our Seldonian classification algorithm. Extending their proof to our setting involves the following minor changes:

1. Changes related to the output of the function used to calculate the utility of a solution: Metevier et al. (2019) consider a utility function that returns the sample reward of a policy. Instead, our utility function (Algorithm 2) outputs the sample loss of a model.

2. Changes due to the form of the fairness constraints: The form of our delayed-impact constraint differs from the more general form of the fairness constraints considered by Metevier et al. (2019). This results in a simplified argument that our algorithm is consistent.

Rather than reword their proof with these minor changes, below we provide their proof with these minor changes incorporated.

> **Theorem 3:** If Assumptions 1–8 hold, then $\lim_{n \to \infty} \Pr(a(D) \neq \texttt{NSF}, \ g(a(D)) \leq 0) = 1$.

We begin by providing definitions and assumptions necessary for presenting our main result. To simplify notation, we assume that there exists only a single delayed-impact constraint and note that the extension of this proof to multiple delayed-impact constraints is straightforward.

Recall that the logged data, $D$, is a random variable. To further formalize this notion, let $(\Omega, \Sigma, p)$ be a probability space on which all relevant random variables are defined, and let $D_n : \Omega \to \mathcal{D}$ be a random variable, where $\mathcal{D}$ is the set of all possible datasets and $D_n = D_c \cup D_f$. We will discuss convergence as $n \to \infty$. $D_n(\omega)$ is a particular sample of the entire set of logged data with $n$ data points, where $\omega \in \Omega$.

**Definition 1** (Piecewise Lipschitz continuous). *We say that a function $f : M \to \mathbb{R}$ on a metric space $(M, d)$ is piecewise Lipschitz continuous with Lipschitz constant $K$ and with respect to a countable partition, $\{M_1, M_2, ...\}$, of $M$ if $f$ is Lipschitz continuous with Lipschitz constant $K$ on all metric spaces in $\{(M_i, d)\}_{i=1}^{\infty}$.*

**Definition 2** ($\delta$-covering). *If $(M, d)$ is a metric space, a set $X \subseteq M$ is a $\delta$-covering of $(M, d)$ if and only if $\max_{y \in M} \min_{x \in X} d(x, y) \leq \delta$.*

Let $\hat{c}(\theta, D_c)$ denote the output of a call to Algorithm 2, and let $c(\theta) := \ell_{\max} + g(\theta)$. The next assumption ensures that $c$ and $\hat{c}$ are piecewise Lipschitz continuous. Notice that the $\delta$-covering requirement is straightforwardly satisfied if $\Theta$ is countable or $\Theta \subseteq \mathbb{R}^m$ for any positive natural number $m$.

**Assumption 6.** *The feasible set of policies, $\Theta$, is equipped with a metric, $d_\Theta$, such that for all $D_c(\omega)$ there exist countable partitions of $\Theta$, $\Theta^c = \{\Theta_1^c, \Theta_2^c, ...\}$, and $\Theta^{\hat{c}} = \{\Theta_1^{\hat{c}}, \Theta_2^{\hat{c}}, ...\}$, where $c(\cdot)$ and $\hat{c}(\cdot, D_c(\omega))$ are piecewise Lipschitz continuous with respect to $\Theta^c$ and $\Theta^{\hat{c}}$ respectively with Lipschitz constants $K$ and $\hat{K}$. Furthermore, for all $i \in \mathbb{N}_{>0}$ and all $\delta > 0$ there exist countable $\delta$-covers of $\Theta_i^c$ and $\Theta_i^{\hat{c}}$.*

Intuitively, Assumption 6 states that *(1)* the cost function used to evaluate classifiers is smooth: similar classifiers have similar costs/performances; and *(2)* each classifier can be described by a set of real-valued parameters, as is the case with all parametric supervised learning algorithms.

Next, we assume that a fair solution, $\theta^\star$, exists such that $g(\theta^\star)$ is not precisely on the boundary of fair and unfair. This can be satisfied by solutions that are arbitrarily close to the fair-unfair boundary.

**Assumption 7.** *There exists an $\epsilon > \xi$ and a $\theta^\star \in \Theta$ such that $g(\theta^\star) \leq -\epsilon$.*

Intuitively, Assumption 7 states that the space of classifiers is not degenerate: at least one fair solution exists such that if we perturb its parameters infinitesimally, it would not become arbitrarily unfair. Next, we assume that the sample loss, $\hat{\ell}$, converges almost surely to $\ell$, the actual expected loss.

**Assumption 8.** $\forall \theta \in \Theta, \hat{\ell}(\theta, D_c) \xrightarrow{a.s.} \ell(\theta)$.

Intuitively, Assumption 8 states that the sample performance of a classifier converges to its true expected performance given enough data. This is similar to the usual assumption, e.g., in the regression setting, that a model's sample Mean Squared Error (MSE) converges to its true MSE given sufficient examples.

We prove Theorem 3 by building up properties that culminate with the desired result, starting with a variant of the strong law of large numbers:

**Property 3** (Khintchine Strong Law of Large Numbers). *Let $\{X_\iota\}_{i=1}^\infty$ be independent and identically distributed random variables. Then $(\frac{1}{n}\sum_{i=1}^n X_\iota)_{n=1}^\infty$ is a sequence of random variables that converges almost surely to $\mathbf{E}[X_1]$, if $\mathbf{E}[X_1]$ exists, i.e., $\frac{1}{n}\sum_{i=1}^n X_\iota \xrightarrow{a.s.} \mathbf{E}[X_1]$.*

*Proof.* See Theorem 2.3.13 of Sen & Singer (1993). ☐

Next, we show that the average of the estimates of $g(\theta)$ converge almost surely to $g(\theta)$:

**Property 4.** *If Assumptions 1, 2, and 4 hold, then $\forall \theta \in \Theta, \mathrm{Avg}(\hat{g}) \xrightarrow{a.s.} g(\theta)$.*

*Proof.* Recall that given Assumptions 1, 2, and 4, Lemmas 1 and 2 hold, i.e., estimates in $\hat{g}$ are i.i.d., and each estimate in $\hat{g}$ is an unbiased estimate of $g(\theta)$. Also, recall that if $n_{\hat{g}}$ is the number of elements in $\hat{g}$, $\mathrm{Avg}(\hat{g}) := \frac{1}{n_{\hat{g}}}\sum_{i=1}^{n_{\hat{g}}} \hat{g}_i$. Then, by Property 3 we have that $\mathrm{Avg}(\hat{g}) \xrightarrow{a.s.} g(\theta)$. ☐

In this proof, we consider the set $\bar{\Theta} \subseteq \Theta$, which contains all solutions that are not fair, and some that are fair but fall beneath a certain threshold: $\bar{\Theta} := \{\theta \in \Theta : g(\theta) > -\xi/2\}$. At a high level, we will show that the probability that the candidate solution, $\theta_c$, viewed as a random variable that depends on the candidate data set $D_c$, satisfies $\theta_c \notin \bar{\Theta}$ converges to one as $n \to \infty$, and then that the probability that $\theta_c$ is returned also converges to one as $n \to \infty$.

First, we will show that the upper bounds $U^+$ (constructed in candidate selection, i.e., Algorithm 2) and $U$ (constructed in the fairness test, i.e., Algorithm 1) converge to $g(\theta)$ for all $\theta \in \Theta$. To clarify notation, we write $U^+(\theta, D_c)$ and $U(\theta, D_f)$ to emphasize that each depends on $\theta$ and the datasets $D_c$ and $D_f$, respectively.

**Property 5.** *If Assumptions 1, 2, 4, and 5 hold, then for all $\theta \in \Theta$, $U^+(\theta, D_c) \xrightarrow{a.s.} g(\theta)$ and $U(\theta, D_f) \xrightarrow{a.s.} g(\theta)$.*

*Proof.* Given Assumption 5, Hoeffding's inequality and Student's $t$-test construct high-confidence upper bounds on the mean by starting with the sample mean of the unbiased estimates (in our case, $\mathrm{Avg}(\hat{g})$) and then adding an additional term (a constant in the case of Hoeffding's inequality). Thus, $U(\theta, D_f)$ can be written as $\mathrm{Avg}(\hat{g}) + Z_n$, where $Z_n$ is a sequence of random variables that converges

(surely for Hoeffding's inequality, almost surely for Student's $t$-test) to zero. So, $Z_n \xrightarrow{\text{a.s.}} 0$, and we need only show that $\text{Avg}(\hat{g}) \xrightarrow{\text{a.s.}} g(\theta)$, which follows from Assumptions 1, 2, and Property 4. We therefore have that $U \xrightarrow{\text{a.s.}} g(\theta)$.

The same argument can be used when substituting $U^+(\theta, D_c)$ for $U(\theta, D_f)$. Notice that the only difference between the method used to construct confidence intervals in the fairness test (that is, $U^+$) and in Algorithm 2 (that is, $U$) is the multiplication of $Z_n$ by a constant $\lambda$. This still results in a sequence of random variables that converges (almost surely for Student's $t$-test) to zero. □

Recall that we define $\hat{c}(\theta, D_c)$ to be the output of Algorithm 2. Below, we show that given a fair solution $\theta^\star$ and data $D_c$, $\hat{c}(\theta^\star, D_c)$ converges almost surely to $\ell(\theta^\star)$, the expected loss of $\theta^\star$.

**Property 6.** *If Assumptions 1, 2, 4, 5, 7, and 8 hold, $\hat{c}(\theta^\star, D_c) \xrightarrow{a.s.} \ell(\theta^\star)$.*

*Proof.* By Property 5 (which holds given Assumptions 1, 2, 4, and 5), we have that $U^+(\theta^\star) \xrightarrow{\text{a.s.}} g(\theta^\star)$. By Assumption 7, we have that $g(\theta^\star) \leq -\epsilon$. Now, let

$$A = \{\omega \in \Omega : \lim_{n \to \infty} U^+(\theta^\star, D_c(\omega)) = g(\theta^\star)\}.$$

Recall that $U^+(\theta^\star, D_c) \xrightarrow{\text{a.s.}} g(\theta^\star)$ means that $\Pr(\lim_{n \to \infty} U^+(\theta^\star, D_c) = g(\theta^\star)) = 1$. So, $\omega$ is in $A$ almost surely, i.e., $\Pr(\omega \in A) = 1$. Consider any $\omega \in A$. From the definition of a limit and the previously established property that $g(\theta^\star) \leq -\epsilon$, we have that there exists an $n_0$ such that for all $n \geq n_0$, Algorithm 2 will return $\hat{\ell}(\theta^\star, D_c)$ (this avoids the discontinuity of the `if` statement in Algorithm 2 for values smaller than $n_0$).

Furthermore, we have from Assumption 8 that $\hat{\ell}(\theta^\star, D_c) \xrightarrow{\text{a.s.}} \ell(\theta^\star)$. Let

$$B = \{\omega \in \Omega : \lim_{n \to \infty} \hat{\ell}(\theta^\star, D_c(\omega)) = \ell(\theta^\star)\}.$$

From Assumption 8, we have that $\omega$ is in $B$ almost surely, i.e., $\Pr(\omega \in B) = 1$, and thus by the countable additivity of probability measures, $\Pr(\omega \in (A \cap B)) = 1$.

Consider now any $\omega \in (A \cap B)$. We have that for sufficiently large $n$, Algorithm 2 will return $\hat{\ell}(\theta^\star, D_c)$ (since $\omega \in A$), and further that $\hat{\ell}(\theta^\star, D_c) \to \ell(\theta^\star)$ (since $\omega \in B$). Thus, for all $\omega \in (A \cap B)$, the output of Algorithm 2 converges to $\ell(\theta^\star)$, i.e., $\hat{c}(\theta^\star, D_c(\omega)) \to \ell(\theta^\star)$. Since $\Pr(\omega \in (A \cap B)) = 1$, we conclude that $\hat{c}(\theta^\star, D_c(\omega)) \xrightarrow{\text{a.s.}} \ell(\theta^\star)$. □

We have now established that the output of Algorithm 2 converges almost surely to $\ell(\theta^\star)$ for the $\theta^\star$ assumed to exist in Assumption 7. We now establish a similar result for all $\theta \in \bar{\Theta}$—that the output of Algorithm 2 converges almost surely to $c(\theta)$ (recall that $c(\theta)$ is defined as $\ell_{\max} + g(\theta)$).

**Property 7.** *If Assumptions 1, 2, 4, and 5 hold, then for all $\theta \in \bar{\Theta}$, $\hat{c}(\theta, D_c) \xrightarrow{a.s.} c(\theta)$.*

*Proof.* By Property 5 (which holds given Assumptions 1, 2, 4, and 5), we have that $U^+(\theta, D_c) \xrightarrow{\text{a.s.}} g(\theta)$. If $\theta \in \bar{\Theta}$, then we have that $g(\theta) > -\xi/2$. We now change the definition of the set $A$ from its definition in the previous property to a similar definition suited to this property. That is, let:

$$A = \{\omega \in \Omega : \lim_{n \to \infty} U^+(\theta, D_c(\omega)) = g(\theta)\}.$$

Recall that $U^+(\theta, D_c) \xrightarrow{\text{a.s.}} g(\theta)$ means that $\Pr(\lim_{n \to \infty} U^+(\theta, D_c) = g(\theta)) = 1$. So, $\omega$ is in $A$ almost surely, i.e., $\Pr(\omega \in A) = 1$. Consider any $\omega \in A$. From the definition of a limit and the previously established property that $g(\theta) > -\xi/2$, we have that there exists an $n_0$ such that for all $n \geq n_0$ Algorithm 2 will return $\ell_{\max} + U^+(\theta, D_c(\omega))$. By Property 5 (which holds given Assumptions 1, 2, 4, and 5), $U^+(\theta, D_c(\omega)) \xrightarrow{\text{a.s.}} g(\theta)$. So, for all $\omega \in A$, the output of Algorithm 2 converges almost surely to $\ell_{\max} + g(\theta)$; that is, $\hat{c}(\theta, D_c(\omega)) \xrightarrow{\text{a.s.}} \ell_{\max} + g(\theta)$, and since $c(\theta) = \ell_{\max} + g(\theta)$, we therefore conclude that $\hat{c}(\theta, D_c(\omega)) \xrightarrow{\text{a.s.}} c(\theta)$. □

By Property 7 and one of the common definitions of almost sure convergence,

$$\forall \theta \in \bar{\Theta}, \forall \epsilon > 0, \Pr \left( \lim_{n \to \infty} \inf\{\omega \in \Omega : |\hat{c}(\theta, D_n(\omega)) - c(\theta)| < \epsilon\} \right) = 1.$$

Because $\Theta$ is not countable, it is not immediately clear that all $\theta \in \bar{\Theta}$ converge simultaneously to their respective $c(\theta)$. We show next that this is the case due to our smoothness assumptions.

**Property 8.** *If Assumptions 1, 2, 4, 5, and 6 hold, then $\forall \epsilon' > 0$,*

$$\Pr \left( \lim_{n \to \infty} \inf\{\omega \in \Omega : \forall \theta \in \bar{\Theta}, |\hat{c}(\theta, D_c(\omega)) - c(\theta)| < \epsilon'\} \right) = 1.$$

*Proof.* Let $C(\delta)$ denote the union of all the points in the $\delta$-covers of the countable partitions of $\Theta$ assumed to exist by Assumption 6. Since the partitions are countable and the $\delta$-covers for each region are assumed to be countable, we have that $C(\delta)$ is countable for all $\delta$. Then by Property 7 (which holds given Assumptions 1, 2, 4, and 5), for all $\delta$, we have convergence for all $\theta \in C(\delta)$ simultaneously:

$$\forall \delta > 0, \forall \epsilon > 0, \Pr \left( \lim_{n \to \infty} \inf\{\omega \in \Omega : \forall \theta \in C(\delta), |\hat{c}(\theta, D_c(\omega)) - c(\theta)| < \epsilon\} \right) = 1. \quad (13)$$

Now, consider a $\theta \notin C(\delta)$. By Definition 2 and Assumption 6, $\exists \theta' \in \bar{\Theta}_i^c, d(\theta, \theta') \leq \delta$. Moreover, because $c$ and $\hat{c}$ are Lipschitz continuous on $\bar{\Theta}_i^c$ and $\bar{\Theta}_i^{\hat{c}}$ (by Assumption 6) respectively, we have that $|c(\theta) - c(\theta')| \leq K\delta$ and $|\hat{c}(\theta, D_c(\omega)) - \hat{c}(\theta', D_c(\omega))| \leq \hat{K}\delta$. So, $|\hat{c}(\theta, D_c(\omega)) - c(\theta)| \leq |\hat{c}(\theta, D_c(\omega)) - c(\theta')| + K\delta \leq |\hat{c}(\theta', D_c(\omega)) - c(\theta')| + \delta(K + \hat{K})$. This means that for all $\delta > 0$:

$$\left( \forall \theta \in C(\delta), |\hat{c}(\theta, D_c(\omega)) - c(\theta)| < \epsilon \right) \implies \left( \forall \theta \in \bar{\Theta}, |\hat{c}(\theta, D_c(\omega)) - c(\theta)| < \epsilon + \delta(K + \hat{K}) \right).$$

Substituting this into (13), we get:

$$\forall \delta > 0, \forall \epsilon > 0, \Pr \left( \lim_{n \to \infty} \inf\{\omega \in \Omega : \forall \theta \in \bar{\Theta}, |\hat{c}(\theta, D_c(\omega)) - c(\theta)| < \epsilon + \delta(K + \hat{K})\} \right) = 1.$$

Now, let $\delta := \epsilon/(K + \hat{K})$ and $\epsilon' = 2\epsilon$. Thus, we have the following:

$$\forall \epsilon' > 0, \Pr \left( \lim_{n \to \infty} \inf\{\omega \in \Omega : \forall \theta \in \bar{\Theta}, |\hat{c}(\theta, D_c(\omega)) - c(\theta)| < \epsilon'\} \right) = 1.$$

$\square$

So, given the appropriate assumptions, for all $\theta \in \bar{\Theta}$, we have that $\hat{c}(\theta, D_c(\omega)) \xrightarrow{\text{a.s.}} c(\theta)$ and that $\hat{c}(\theta^\star, D_c(\omega)) \xrightarrow{\text{a.s.}} \ell(\theta^\star)$. Due to the countable additivity property of probability measures and Property 8, we have the following:

$$\Pr \left( \left[ \forall \theta \in \bar{\Theta}, \lim_{n \to \infty} \hat{c}(\theta, D_c(\omega)) = c(\theta) \right], \left[ \lim_{n \to \infty} \hat{c}(\theta^\star, D_c(\omega)) = \ell(\theta^\star) \right] \right) = 1, \quad (14)$$

where $\Pr(A, B)$ denotes the joint probability of $A$ and $B$.

Let $H$ denote the set of $\omega \in \Omega$ such that (14) is satisfied. Note that $\ell_{\max}$ is defined as the value always greater than $\ell(\theta)$ for all $\theta \in \Theta$, and $g(\theta) \geq -\xi$ for all $\theta \in \bar{\Theta}$. So, for all $\omega \in H$, for sufficiently large $n$, candidate selection will not define $\theta_c$ to be in $\bar{\Theta}$. Since $\omega$ is in $H$ almost surely ($\Pr(\omega \in H) = 1$), we therefore have that $\lim_{n \to \infty} \Pr(\theta_c \notin \bar{\Theta}) = 1$.

The remaining challenge is to establish that, given $\theta_c \notin \bar{\Theta}$, the probability that the fairness test returns $\theta_c$ rather than NSF converges to one as $n \to \infty$. By Property 5, we have that $U(\theta_c, D_f) \xrightarrow{\text{a.s.}} g(\theta_c)$. Furthermore, by the definition of $\bar{\Theta}$, when $\theta_c \notin \bar{\Theta}$ we have that $g(\theta_c) < -\xi/2$. So, $U(\theta_c, D_f)$ converges almost surely to a value less than $-\xi/2$. Since the fairness test returns $\theta_c$ rather than NSF if $U(\theta_c, D_f) \leq -\xi/4$ and $U(\theta_c, D_f)$ converges almost surely to a value less than $-\xi/2$, it follows that the probability that $U(\theta_c, D_f) \leq -\xi/4$ converges to one. Hence, given that $\theta_c \notin \bar{\Theta}$, the probability that $\theta_c$ is returned rather than NSF converges to one.

We therefore have that **1)** the probability that $\theta_c \notin \bar{\Theta}$ converges to one as $n \to \infty$ and **2)** given that $\theta_c \notin \bar{\Theta}$, the probability that $\theta_c$ is returned rather than NSF converges to one. Since $\theta_c \notin \bar{\Theta}$ implies that $\theta_c$ is fair, these two properties imply that the probability that a fair solution is returned converges to one as $n \to \infty$.

# H  A DISCUSSION ON THE INTUITION AND IMPLICATIONS OF OUR ASSUMPTIONS

The theoretical results in this paper, which ensure ELF's convergence and high-confidence fairness guarantees, are based on Assumptions 1–8. In this section, we provide an intuitive, high-level discussion on the meaning and implications of these assumptions. Our goal is to show that they are standard in the machine learning literature and reasonable in many real-life settings.

**Assumption 1.** This assumes the commonly-occurring setting in which a classifier's prediction depends only on an input feature vector, $X$. It is a formal characterization of the classic supervised learning setting; that is, that machine learning models should predict the target variable of interest, $Y$, based only on an input feature vector, $X$. In other words, we are dealing with a standard classification problem.

**Assumption 2.** While the delayed impact observation, $I$, depends on the prediction, $\hat{Y}$, it does not depend on *how* the prediction was made. For example, it makes no difference whether a neural network or a support vector machine made a loan-repayment prediction.

**Assumptions 3 and 4.** These assumptions can be trivially satisfied when our algorithm uses standard modern stochastic classifiers that place non-zero probability on all outputs; for instance, when using Softmax layers in a neural network. This assumption is common in the offline RL literature—for example, in methods that evaluate new policies given information from previously-deployed policies.

**Assumption 5.** Our algorithm uses standard statistical tools, common in the machine learning literature, to compute confidence bounds: Hoeffding's inequality and Student's t-test. Hoeffding's inequality can be applied under mild assumptions. In our running example, lending decisions made by a bank may have a delayed impact, e.g., on an applicant's future savings rate. Hoeffding's inequality holds if the bank knows the minimum and maximum savings rate possible (i.e., the DI is bounded in some interval $[a, b]$). Bounds produced by Student's t-test hold exactly if the sample mean is normally distributed, and in the limit (as the number of samples increases) if the sample mean follows a different distribution. With few samples, bounds based on Student's t-test may hold approximately. Despite this, their use remains effective and commonplace in the sciences, including, e.g., in high-risk medical research (Thomas et al., 2019).

**Assumption 6.** The cost function used to evaluate classifiers is smooth: similar classifiers have similar costs/performances. Smoothness assumptions of this type are common the machine learning literature. Also, each classifier can be described by a set of real-valued parameters ($\theta \in \mathbb{R}^m$), as is the case with all parametric supervised learning algorithms.

**Assumption 7.** The space of classifiers is not degenerate: at least one fair solution exists such that if we perturb its parameters infinitesimally, it would not become arbitrarily unfair.

**Assumption 8.** The sample performance of a classifier converges to its true expected performance given enough data. This is similar to the usual assumption, e.g., in the regression setting, that a model's sample Mean Squared Error (MSE) converges to its true MSE given sufficient examples.

# I  FULL ALGORITHM

Algorithm 2 presents the cost function used in candidate selection (line 3 of Algorithm 1), where a strategy like the one used in the fairness test is used to calculate the cost, or utility, of a potential solution $\theta$. Again, unbiased estimates of $g(\theta)$ are calculated, this time using dataset $D_c$ (lines 2–4). Instead of calculating a high-confidence upper bound on $g(\theta)$ using $U_{\texttt{Hoeff}}$ or $U_{\texttt{ttest}}$, we calculate an *inflated* upper bound $U^+$. Specifically, we inflate the width of the confidence interval used to compute the upper bound (lines 5–10). This is to mitigate the fact that multiple comparisons are performed on the same dataset ($D_c$) during the search for a candidate solution (see line 3 of Algorithm 1), which often leads candidate selection to overestimate its confidence that the solution it picks will pass the fairness test. Our choice to inflate the confidence interval in this way, i.e., considering the size of the dataset $D_f$ used in the fairness test and the use of scaling constant $\lambda$, is empirically driven and was first proposed for other Seldonian algorithms (Thomas et al., 2019).

---

**Algorithm 2** `cost`

---

**Input**: **1)** the vector $\theta$ that parameterizes model $\pi$; **2)** $D_c = \{(X_i, Y_i, T_i, \widehat{Y}_i^\beta, I_i^\beta)\}_{i=1}^m$; **3)** confidence level $\delta$; **4)** tolerance value $\tau$; **5)** the behavior model $\beta$; **6)** $\texttt{Bound} \in \{\texttt{Hoeff}, \texttt{ttest}\}$; and **7)** the number of data points in $D_f$, denoted $n_{D_f}$.
**Output**: The cost of $\pi$.

1: $\hat{g} \leftarrow \langle \, \rangle$
2: **for** $i \in \{1, ..., m\}$ **do**
3:    **if** $c(X_i, Y_i, T_i)$ is $\texttt{True}$ **then** $\hat{g}$.append$\left(\tau - \frac{\pi_\theta(X_i, \widehat{Y}_i^\beta)}{\beta(X_i, \widehat{Y}_i^\beta)} I_i^\beta\right)$ **end if**
4: **end for**
5: Let $\lambda = 2$;   $n_{\hat{g}} = \texttt{length}(\hat{g})$
6: **if** $\texttt{Bound}$ is $\texttt{Hoeff}$ **then**
7:    $a, b \leftarrow$ upper and lower bounds of $g$
8:    $U^+ = \frac{1}{n_{\hat{g}}} \left(\sum_{\iota=1}^{n_{\hat{g}}} \hat{g}_\iota\right) + \lambda(b-a)\sqrt{\frac{\log(1/\delta)}{(2n_{D_f})}}$
9: **else if** $\texttt{Bound}$ is $\texttt{ttest}$ **then** $U^+ = \frac{1}{n_{\hat{g}}} \left(\sum_{\iota=1}^{n_{\hat{g}}} \hat{g}_\iota\right) + \lambda\frac{\sigma(\hat{g})}{\sqrt{n_{D_f}}} t_{1-\delta, n_{D_f}-1}$
10: **end if**
11: $\ell_{\max} = \max_{\theta' \in \Theta} \hat{\ell}(\theta', D_c)$
12: **if** $U^+ \leq -\frac{\xi}{4}$ **return** $\hat{\ell}(\theta, D_c)$ **else return** $(\ell_{\max} + U^+)$

---

**Algorithm 3** `ELF` with Multiple Constraints

---

**Input**: **1)** dataset $D = \{(X_i, Y_i, T_i, \widehat{Y}_i^\beta, I_i^\beta)\}_{i=1}^n$; **2)** the number of delayed-impact constraints we wish to satisfy, $k$; **3)** a sequence of Boolean conditionals $(c_j)_{j=1}^k$ such that for $j \in \{1, ..., k\}$, $c_j(X_i, Y_i, T_i)$ indicates whether the event associated with the data point $(X_i, Y_i, T_i, \widehat{Y}_i^\beta, I_i^\beta)$ occurs; **4)** confidence levels $\delta = (\delta_j)_{j=1}^k$, where each $\delta_j \in (0, 1)$ corresponds to delayed-impact constraint $g_j$; **5)** tolerance values $\tau = (\tau_j)_{j=1}^k$, where each $\tau_j$ is the tolerance associated with delayed-impact constraint $g_j$; **6)** the behavior model $\beta$; and **7)** an argument $\texttt{Bound} \in \{\texttt{Hoeff}, \texttt{ttest}\}$ indicating which method for calculating upper bounds to use.
**Output**: Solution $\theta_c$ or $\texttt{NSF}$.

1: $D_c, D_f \leftarrow \texttt{partition}(D)$
2: $\theta_c \leftarrow \arg\min_{\theta \in \Theta} \texttt{cost}(\theta, D_c, k, \delta, \tau, \beta, \texttt{Bound}, \texttt{length}(D_f))$
3: $U \leftarrow \langle \, \rangle$
4: **for** $j \in \{1, ..., k\}$ **do**
5:    $\hat{g}_j \leftarrow \langle \, \rangle$
6:    **for** $i \in \{1, ..., n\}$ **do**
7:      **if** $c_j(X_i, Y_i, T_i)$ is $\texttt{True}$ **then** $\hat{g}_j$.append$\left(\tau_j - \frac{\pi_{\theta_c}(X_i, \widehat{Y}_i^\beta)}{\beta(X_i, \widehat{Y}_i^\beta)} I_i^\beta\right)$ **end if**
8:    **end for**
9:    **if** $\texttt{Bound}$ is $\texttt{Hoeff}$ **then** $U$.append$(U_{\texttt{Hoeff}}(\hat{g}_j))$ **else** $U$.append$(U_{\texttt{ttest}}(\hat{g}_j))$ **end**
10: **end for**
11: **if** $\forall j \in \{1, ..., k\}, U_j \leq 0$ **then return** $\theta_c$ **else return** $\texttt{NSF}$

---

If $U^+ \leq -\xi/4$, a small negative constant, the cost associated with the loss of $\theta$ is returned. Otherwise, the cost of $\theta$ is defined as the sum of $U^+$ and the maximum loss that can be obtained on dataset $D_c$ (lines 11–12). This discourages candidate selection from returning models unlikely to pass the fairness test. We consider $-\xi/4$, instead of $0$ as the fairness threshold in $\texttt{ELF}$ to ensure consistency. This is discussed in more detail in Appendix G.

Algorithm 3 shows $\texttt{ELF}$ with multiple constraints. The changes relative to Algorithm 1 are relatively small: instead of considering only a single constraint, the fairness test loops over all $k$ constraints and only returns the candidate solution if all $k$ high-confidence upper bounds are at most zero. Similarly, the cost function, Algorithm 4, changes relative to Algorithm 2 in that when predicting the outcome of the fairness test it includes this same loop over all $k$ delayed-impact constraints.

---

**Algorithm 4** `cost` with Multiple Constraints

---

**Input**: **1)** the vector $\theta$ that parameterizes a classification model $\pi$; **2)** candidate dataset $D_c = \{(X_i, Y_i, T_i, \widehat{Y}_i^\beta, I_i^\beta)\}_{i=1}^m$; **3)** the number of delayed-impact constraints we wish to satisfy, $k$; **4)** a sequence of Boolean conditionals $(c_j)_{j=1}^k$ such that for $j \in \{1, ..., k\}$, $c_j(X_i, Y_i, T_i)$ indicates whether the event associated with the data point $(X_i, Y_i, T_i, \widehat{Y}_i^\beta, I_i^\beta)$ occurs; **5)** confidence levels $\delta = \{\delta_j\}_{j=1}^k$, where each $\delta_j \in (0, 1)$ corresponds with constraint $g_j$; **6)** tolerance values $\tau = \{\tau_j\}_{j=1}^k$, where each $\tau_j$ is the tolerance associated with delayed-impact constraint $g_j$; **7)** the behavior model $\beta$; **8)** an argument `Bound` $\in \{$`Hoeff`, `ttest`$\}$ indicating which method for calculating upper bounds to use; and **9)** the number of data points in dataset $D_f$, denoted $n_{D_f}$.

**Output**: The cost associated with classification model $\pi_\theta$.

1: **for** $j \in \{1, ..., k\}$ **do**
2:      $\hat{g}_j \leftarrow \langle \, \rangle$
3:      **for** $i \in \{1, ..., m\}$ **do**
4:          **if** $c_j(X_i, Y_i, T_i)$ is `True` **then** $\hat{g}_j$.append$\left(\tau_j - \frac{\pi_\theta(X_i, \widehat{Y}_i^\beta)}{\beta(X_i, \widehat{Y}_i^\beta)} I_i^\beta\right)$ **end if**
5:      **end for**
6:      Let $\lambda = 2$;    $n_{\hat{g}_j} = $ `length`$(\hat{g}_j)$
7:      **if** `Bound` is `Hoeff` **then**
8:          Let $a, b$ be the lower and upper bounds of $g_j$
9:          $U_j^+ = \frac{1}{n_{\hat{g}_j}} \left(\sum_{\iota=1}^{n_{\hat{g}_j}} (\hat{g}_j)_\iota\right) + \lambda(b-a)\sqrt{\frac{\log(1/\delta_j)}{(2n_{D_f})}}$
10:      **else if** `Bound` is `ttest` **then**
11:          $U_j^+ = \frac{1}{n_{\hat{g}_j}} \left(\sum_{\iota=1}^{n_{\hat{g}_j}} (\hat{g}_j)_\iota\right) + \lambda\frac{\sigma(\hat{g}_j)}{\sqrt{n_{D_f}}} t_{1-\delta_j, n_{D_f}-1}$
12:      **end if**
13: **end for**
14: $\ell_{\max} = \max_{\theta' \in \Theta} \hat{\ell}(\theta', D_c)$
15: **if** $\forall j \in \{1, ..., k\}, U_j^+ \leq -\xi/4$ **then return** $\hat{\ell}(\theta, D_c)$ **else return** $\left(\ell_{\max} + \sum_{j=1}^k U_j^{\text{inflated}}\right)$

---

## J  OTHER EXPERIMENTS

In our experiments, we consider a classifier that predicts whether youth in foster care will have a job after leaving the program. (4) indicates that if they are predicted to get a job, they are more likely to receive financial aid, which may lead to a positive (higher) delayed impact observation—$I_i^\psi$ increases when $\widehat{Y}_i^\psi = 1$. Whether DI is strongly affected by a classifier's predictions is regulated by $\alpha$. DI is also affected by Gaussian noise representing stochasticity in the environment. Different means and variances indicate that the DI on youth of different races may (due to social biases) differ. Changing the noise distribution would affect how youth of different races are affected by predictions, thus modeling a different type of society.

In all experiments, our implementation of `ELF` used CMA-ES (Hansen & Ostermeier, 2001) to search over the space of candidate solutions and the `ttest` concentration inequality. We partitioned the dataset $D$ into $D_c$ and $D_f$ using a stratified sampling approach where $D_c$ contains 60% of the data and $D_f$ contains 40% of the data.

In this section, we present the complete set of results for RQ1 and RQ2: does `ELF` enforce DI constraints, with high probability, when existing fairness-aware algorithms often fail; and what is the cost of enforcing DI constraints. In particular, we show the performance of the five algorithms being compared, in terms of failure rate, probability of returning a solution, and accuracy, for different values of $\alpha$ and as a function of $n$. Notice that Figures 4 and 5 present results consistent with the observations made in Section 5: the qualitative behavior of all considered algorithms remains the same for all values of $\alpha$.

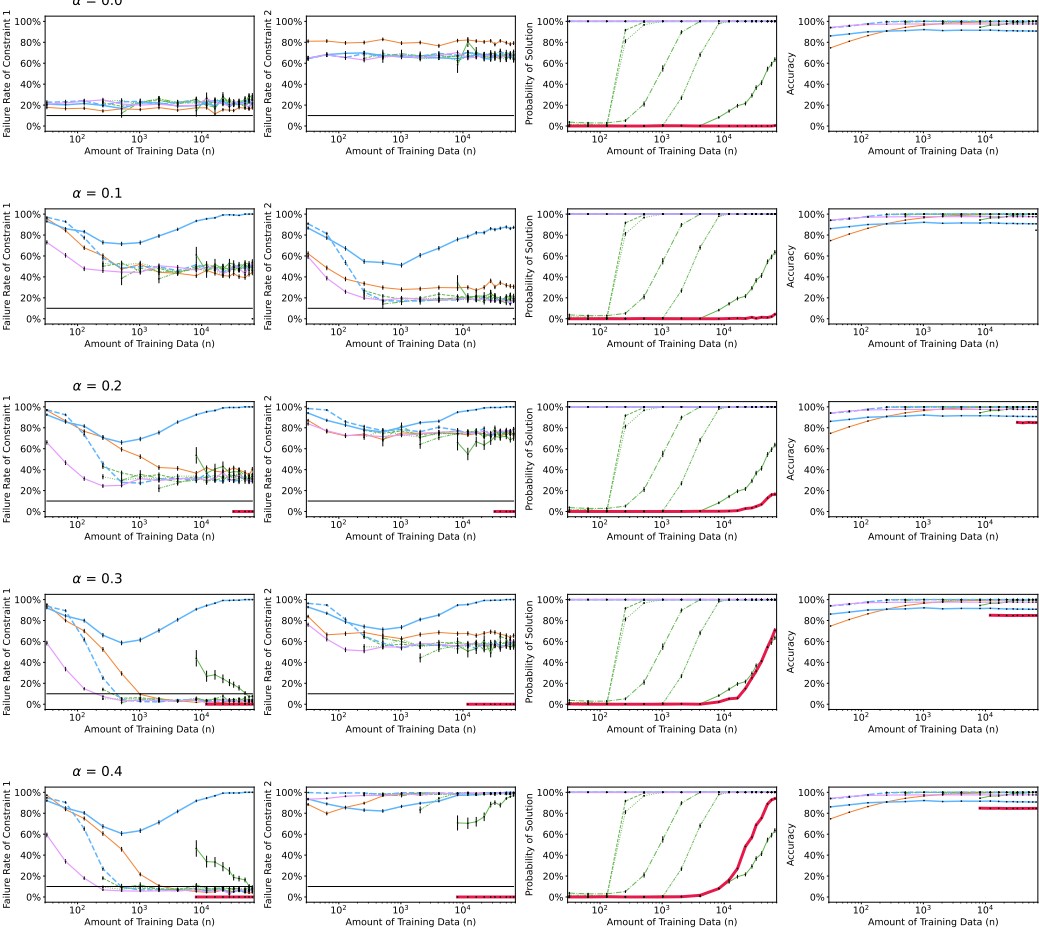

Figure 4: Algorithms' performances in terms of failure rate (leftmost two columns), probability of returning a solution (second from the right column), and accuracy (right column), as a function of $n$ and for different values of $\alpha$. The black horizontal lines indicate the maximum admissible probability of unfairness, $\delta_0 = \delta_1 = 10\%$. All plots use the following legend: ——ELF ——LR --- QSA with DP -·-·QSA with EqOdds -··-··QSA with EqOpp ——QSA with PE ········QSA with DisImp ——Fairlearn with DP --- Fairlearn with EqOdds ——Fairness Constraints.

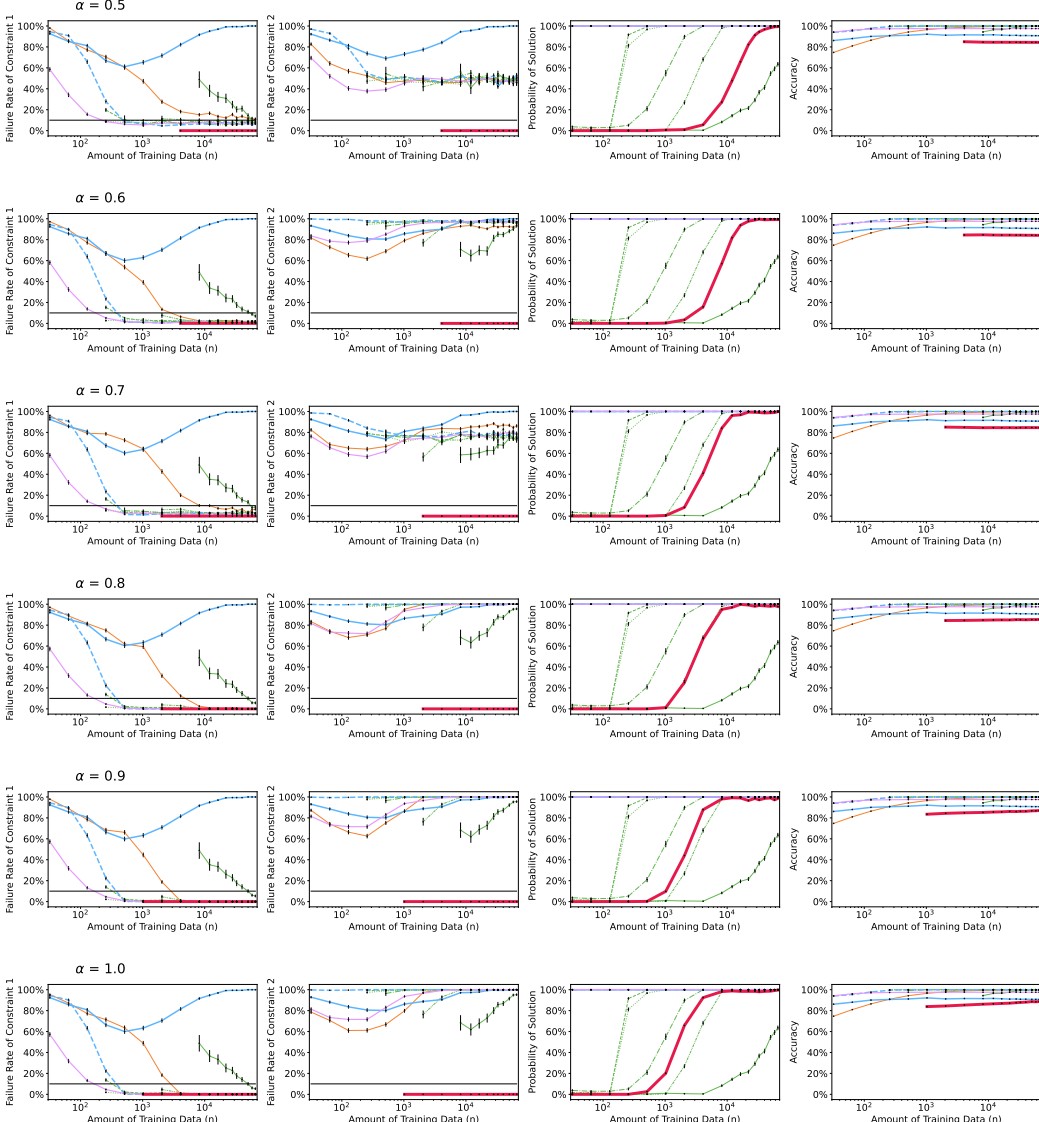

Figure 5: Algorithms' performance in terms of failure rate (leftmost two columns), probability of returning a solution (second from the right column), and accuracy (right column), as a function of $n$ and for different values of $\alpha$. The black horizontal lines indicate the maximum admissible probability of unfairness, $\delta_0 = \delta_1 = 10\%$. All plots use the following legend: —— ELF —— LR - - - QSA with DP -·-·- QSA with EqOdds ······ QSA with EqOpp —— QSA with PE ········ QSA with DisImp —— Fairlearn with DP - - - Fairlearn with EqOdds —— Fairness Constraints.

## K    RELATED WORK

Most prior work on the social implications of ML study static fairness without considering the long-term impact of model decisions (Calders et al., 2009; Zafar et al., 2017; Hardt et al., 2016; Dwork et al., 2012). However, there exists a growing body of work that examines the long-term impact of fairness in ML (D'Amour et al., 2020; Hu & Chen, 2018a;b; Liu et al., 2018; Heidari et al., 2019; Zhang et al., 2020a; Mouzannar et al., 2019). In this paper, we build upon this prior work and present the first method that uses historical data to train a classifier with high-confidence delayed-impact fairness guarantees when the analytic model of the relationship between classifiers' predictions and delayed impact is not known *a priori*.

Wen et al. (2019) and Tang et al. (2020) present work similar to ours. Wen et al. (2019) propose modeling delayed impact using a Markov decision process (MDP) with two different reward functions: one for the decision-maker, e.g., a bank, and another for each individual, e.g., loan applicant. Wen et al. (2019) introduce algorithms that are able to estimate near-optimal policies (in terms of cumulative reward of the decision-maker) while enforcing static fairness constraints (e.g., demographic parity and equal opportunity). Importantly, Wen et al. (2019) introduce a method that ensures that *static* fairness constraints hold for all time during a sequence of decisions. We, by contrast, study the orthogonal problem of ensuring fairness with respect to user-defined *delayed-impact* measures. The method proposed by Tang et al. (2020), unlike ELF, considers the online multi-armed bandit learning setting in which there are no features—the algorithm does not differentiate individuals within a group while making decisions. We, by contrast, tackle the problem of high-confidence fairness in the classification setting.

Work by Ge et al. (2021) and Hu & Zhang (2022) study similar but orthogonal problem settings. Ge et al. (2021) and Hu & Zhang (2022) train classifiers that satisfy static fairness constraints in non-stationary settings (e.g., a recommendation system where a person's interests may change over time). Importantly, both Ge et al. (2021) and Hu & Zhang (2022) require prior knowledge of analytic, accurate models of the environment—for example, in the form of probabilistic graphical models. Our method, by contrast, does not require a model or simulator of the environment, nor prior knowledge about the relationship between a classifier's predictions and the resulting delayed impact.

Zhang et al. (2020b) investigate the long-term effects of repeatedly deploying myopic policies that optimize static fairness constraints. They rely on knowledge of analytical models of "user qualification"—models characterizing whether an individual will be qualified (e.g., likely to repay a loan) at a given time. ELF, by contrast, ensures long-term fairness with high confidence without requiring access to any models. Chi et al. (2022) introduce a new fairness definition in multi-step problems where different demographic groups are represented as different Markov decision processes. The goal of the proposed method is to identify policies that satisfy an alternative type of constraint; in particular, their method's objective is to identify policies that result in return parity between groups. This technique assumes an agent that can repeatedly interact with its environment until fair policies are identified. We, by contrast, investigate an orthogonal setting: ELF addresses the classification setting where a fixed dataset of historical observations is available (based on which a fair solution should be identified with high confidence), but in which the agent/algorithm cannot further interact with its environment to collect more data. Furthermore, ELF was not designed to tackle any one particular type of constraint; it is compatible with a wide range of (user-specified) constraints. D'Amour et al. (2020)'s goal is not to propose a new method; instead, they *evaluate* the DI resulting from a given classifier's predictions, under the assumption that an accurate simulator of the environment is available. By contrast, we propose a new method for *training* classifiers that ensure that DI fairness constraints are satisfied, without requiring accurate simulators.

In another line of work, researchers have shown that the fairness of ML algorithms can be improved by manipulating the training data, rather than the learning algorithm. This goal can be achieved, for example, by removing data that violates fairness properties (Verma et al., 2021) or by inserting data inferred using fairness properties (Salimi et al., 2019). Again, while these methods can improve the fairness of learned models, they were designed to enforce static fairness constraints and do not enforce fairness with respect to the delayed impact resulting from deploying such learned models.

Lastly, this paper introduces a method that extends the existing body of work on *Seldonian algorithms (Thomas et al., 2019)*. Seldonian algorithms provide fairness guarantees with high probability and have been shown to perform well in real-world applications given reasonable amounts of training

data (Thomas et al., 2019; Metevier et al., 2019). They also—by construction—provide a straight-forward way for users to define multiple notions of fairness that can be simultaneously enforced (Thomas et al., 2019). The technique introduced in this paper (`ELF`) is the first supervised-learning Seldonian algorithm capable of providing high-probability fairness guarantees in terms of delayed impact.

