# OpenReview forum: "Enforcing Delayed-Impact Fairness Guarantees"
_ICLR.cc/2023/Conference — Submitted to ICLR 2023_

### Official Review · Reviewer_ABty · 2022-10-21

**Confidence:** 4
**Correctness:** 3
**Technical Novelty And Significance:** 3
**Empirical Novelty And Significance:** 2
**Recommendation:** 5

**Clarity, Quality, Novelty And Reproducibility:**

The paper is clearly written. The setting is novel and I think at least the theoretical results of the paper should be easily reproducible.

**Strength And Weaknesses:**

------------------------------
Strengths
------------------------------
-- The delayed impact of fairness in machine learning has not received much attention in the community (mainly due to the hardness of modeling this impact) so in that sense, it is great to see papers on this topic.

--The paper is pretty well written and it is clear to follow the high-level ideas of the paper.

------------------------------
Weaknesses
------------------------------
--Assumption 3 is strong. While it is common in the RL community to have this assumption, this is less realistic in the algorithmic fairness literature and a possible source of bias. How can this assumption be justified?

-- The theoretical analysis of the paper is rather straightforward.

-- The experimental analysis is not convincing.
(a) None of the baselines considers the delayed impact into consideration. Is it surprising to see that ELF will have a much lower failure rate compared to the baselines?
(b) How did you come up with the form of delayed impact in Equation (4)? In general, having access to such data is a major challenge in the practicality of this work.
(c) The number of data points needed for ELF so that it outputs a solution is very large. Does this problem become less severe when looking at lower-confidence solutions? What is the dependency on the confidence parameter?
(d) The accuracy of the ELF is significantly lower than the baselines.



**Summary Of The Paper:**

The paper studies the delayed impact of fairness in machine learning and provides a new model that models this effect as opposed to the commonly studied static notions of fairness. The paper introduces an algorithm called ELF (Enforcing Long-term Fairness) to provide high confidence delayed impact fairness guarantees (when possible). The authors demonstrate the effectiveness of ELF empirically.


**Summary Of The Review:**

While I think the paper tackles an interesting problem, it is hard to justify the proposed modeling and practicality of the results.

---

> ### Author Response · Authors · 2022-11-15
> **Response/comments to reviewer ABty**
>
> We thank the reviewer for acknowledging the relevance of the problem we tackle: ensuring long-term fairness, with high confidence, when the analytical relationship between predictions and delayed impact is unknown. We appreciate their positive comments on the quality and clarity of our paper.
>
> We appreciate the reviewer's constructive comments. We address their questions below.
>
> ---
>
> **Assumption 3 is strong. How can it be justified/satisfied?**
>
> Assumption 3 is trivially satisfied when our algorithm uses standard stochastic classifiers that place non-zero probability on its outputs; for example, when using any modern neural network classifiers with a Softmax layer, which are commonplace in the supervised learning literature.
>
> Moreover, in applications where full support cannot be ensured, notice that our method's guarantees still hold if delayed-impact observations are always positive or negative. In these cases, as discussed in (Thomas et al., 2015), our importance sampling estimators would result in a lower (or upper) bound, respectively, on the delayed impact resulting from predictions made by a new classifier.
>
> ---
>
> **The theoretical analyses are straightforward.**
>
> We argue that the theoretical analyses shown in our paper are *not* straightforward. We prove three different theoretical results that are not trivial: *(i)* the delayed impact estimator we proposed is unbiased; *(ii)* our method returns, with high confidence, a fair solution; and *(iii)* given enough data, our method is guaranteed to find and return a fair solution if one exists. Concretely, the theoretical results we introduced (Theorems 1, 2, and 3) required approximately 7 pages to be proven.
>
> Moreover, we argue that if a novel solution to an open problem is presented, and can be shown to have strong formal correctness and convergence guarantees via "straightforward analyses", this should be considered a strength, not a weakness. In other words, we argue that scientific contributions should *not* have to be overly complicated to be considered important or valid.
>
> ---
>
> **Comparison with delayed impact-aware baselines.**
>
> As discussed in the paper, to the best of our knowledge, ELF is the *only* existing method in the literature to ensure long-term fairness when analytical models of the prediction-DI relation are unknown. Thus, we compared ELF with the closest fairness-aware methods that do not require a model and depend only on historical data. As discussed in Section 5, currently, *no* other model-free long-term fairness-aware baselines exist with which we could compare.
>
> ---
>
> **On the choice of the form of Equation 4.**
>
> In our experiments, we consider a classifier that predicts whether youth in foster care will have a job after leaving the program. Eq.4 indicates that if they are predicted to get a job ($\widehat Y^\psi_i=1$), they are more likely to receive financial aid, which may lead to a positive (higher) delayed impact. In Eq.4, the delayed impact $I^\psi_i$ increases when $\widehat Y^\psi_i=1$. Whether DI is strongly affected by a classifier's predictions is regulated by $\alpha$. DI is also affected by Gaussian noise representing stochasticity in the environment. Different means and variances indicate that the DI on youth of different races may (due to social biases) differ. A discussion on the form of Eq.4, similar to the one presented above, can be found in Appendix J.
>
> Importantly, notice that Eq.4 is used, in our paper, solely to generate challenging parameterized datasets with different levels of prediction-DI dependency. ELF does *not* have access to this equation, nor depend on/assume its particular form. It only requires (in this particular experiment) access to the data generated by it. In practical applications, users of ELF would not have to design such equations: Eq.4 was designed to model the setting described above so we could perform a thorough set of experiments to test our method's reliability under different conditions. In practice, ELF would rely only on empirical observations of the delayed impact resulting from the deployment of a given classifier, as discussed in the Introduction.

---

> > ### Author Response · Authors · 2022-11-15
> > **Response/comments to reviewer ABty**
> >
> > **Number of instances required for ELF to return a solution.**
> >
> > Figure 2 shows that ELF returns solutions with 91\% probability when given just $n{=}4{,}096$ samples. Datasets with $4{,}000$ instances are considered small in many real-life supervised learning settings.
> >
> > Importantly, notice that as the amount of training data increases, the probability of ELF returning solutions increases rapidly. Although three competing techniques (Fairlearn, Fairness Constraints, and LR) always return solutions, independently of the amount of training data, these solutions ***never*** satisfy both DI constraints. QSA often returns candidate solutions with less training data than ELF; these solutions, however, also fail to satisfy both DI constraints simultaneously.
> >
> > In other words: even though other fairness-aware methods return solutions with fewer instances, they consistently *fail* to return fair solutions—which is the primary objective of this paper.
> >
> > The reviewer is correct that if a particular application requires less confidence that the returned solution is fair (that is, a higher value of $\delta$ is used), then it is likely that the algorithm will return solutions with even fewer training data samples.
> >
> > ---
> >
> > **ELF's accuracy.**
> >
> > The reviewer is correct that there are unavoidable, domain-specific trade-offs between fairness and accuracy. Under low-data regimes ($n{=}1{,}000$), ELF's accuracy is 83\%, while competing methods have accuracy higher than 90\%. Importantly, however, notice that whenever competing methods have higher accuracy than ours, they *consistently return unfair solutions*. Since the objective of this paper is to ensure fairness with high probability, we thus argue that the competing methods implement a *worse* trade-off between fairness and accuracy: they do achieve slightly better accuracy, but at the cost of *consistently* failing to satisfy one or more fairness constraints. ELF, by contrast, ensures that *all* fairness constraints are satisfied} with high probability. Finally, notice that as the amount of available data increases, ELF's accuracy approaches that of the other methods.

---

### Official Review · Reviewer_51vQ · 2022-10-23

**Confidence:** 2
**Correctness:** 3
**Technical Novelty And Significance:** 3
**Empirical Novelty And Significance:** 3
**Recommendation:** 5

**Clarity, Quality, Novelty And Reproducibility:**

The problem formulation is not very clear.


**Strength And Weaknesses:**

+ The setting is interesting where the relationships between the delayed impact and prediction are unknown. A reinforcement learning solution is developed to infer the delayed impact from the observation using importance sampling.
+ Given the inferred delayed impact, a classification is built to ensure DI.
+ The theoretical analysis seems convincing.

- The variable of delayed impact observation and the formulation of long-term fairness is unclear. Eq 1 defines g(theta) for DI but it is not clear why a fair algorithm must ensure Eq 1. Usually, fairness means the absence of disparity between two groups. But variable I is an observation, e.g., the rate. It is unclear why the larger I corresponds to better DI. In brief, DI and DI fairness is not well defined. The confusion makes Sec 2 hard to understand.
- The setting of no analytic relationship between DI and prediction is controversial. Assumption 2 assumes the relationship. Otherwise, the off-policy learning cannot infer the delayed impact. In addition, it is unclear how the performance of importance sampling affects the downstream classification.

**Summary Of The Paper:**

This paper investigates the fair classification problem in the long term. It introduces a new algorithm, ELF, to solve the fair classification problem as both a classification and a reinforcement learning problem — classification for optimizing the main objective (a measure of classification loss) and reinforcement learning when taking DI into account.

**Summary Of The Review:**

This paper presents a solution to classification with long-term fairness constraints. A new setting is considered where the relationship between the delayed impact and prediction is unknown. To handle this challenge, the paper introduces a few assumptions and infers the delayed impact from prediction using a reinforcement learning method. Then the inferred DI is used for classification.

The method seems meaningful but the formulation is not clear. The statements about DI and fairness are confusing. It is not clear why Eq 2<0 is necessary for model fairness.

Assumption 2 is contrary to the setting of the analytic relationship between DI and prediction.

---

> ### Author Response · Authors · 2022-11-15
> **Response/comments to reviewer 51vQ**
>
> We thank the reviewer for their positive comments on the novelty of our approach, which simultaneously formulates the fair classification problem as both a classification and a reinforcement learning problem. Furthermore, we appreciate their comments on the formal guarantees of our algorithm—it is proven to identify nontrivial solutions (fair classifiers) with high confidence.
>
> We thank the reviewer for their constructive comments and suggestions. We address their questions below.
>
> ---
>
> **Why does ensuring that $g(\theta) \leq 0$ imply fairness?**
>
> In our paper, the function $g$ is a delayed impact objective. We adopted the convention that $g(\theta) \leq 0$ implies that the classifier $\theta$ is fair. For example, assume the delayed impact of lending decisions is the savings rate of a loan applicant. Assume that the bank would like to ensure that, with high probability, the savings rate ($I$) resulting from predictions made by a novel classifier will not be lower than some threshold $\tau$. The bank would thus like to ensure that $\mathbb E[I] \geq \tau$. This expression can be rewritten as $\mathbb E[I] - \tau \geq 0$, which implies that $\tau - \mathbb E[I] \leq 0$. In this case, $g$ could be defined as $\tau - \mathbb E[I]$, and then, by construction, whenever $g(\theta) \leq 0$, it follows that the savings rate is no lower than the threshold $\tau$. That is, $g(\theta) \leq 0$ implies that the solution is fair.
>
> Notice that the form of delayed impact objective in Eq.1 allows us to represent many other DI fairness notions studied in the literature, such as Liu et al.'s long-term improvement. In Appendix E, we also show *(1)* how ELF can be used to enforce static fairness constraints, which are common in the literature; *(2)* how ELF can be used to enforce constraints on the minimum performance of the resulting classifier; and *(3)* how ELF is compatible with more general definitions of delayed impact fairness, such ensuring that the delayed impact of a classifier's predictions is approximately equal for loan applicants of group $A$ and group $B$.
>
> ---
>
> **Can fairness be the absence of disparity between two groups?**
>
> We agree that several different fairness definitions exist in the literature—both in the case of static fairness and delayed-impact fairness (for a survey, see Verma and Rubin, 2018). As mentioned above, the delayed impact objective functions considered in this paper can represent many fairness notions studied in the literature, such as Liu et al.'s long-term improvement. Still related to the reviewer's question about fairness as the absence of disparity, we kindly point out our discussion in Appendix E, where we show that ELF can be readily adapted to enforce fairness definitions involving, for instance, the disparity between two groups.
>
> ---
>
> **Why do larger values of $I$ correspond to better DI?**
>
> We adopted a convention that large delayed-impact observations ($I$) correspond to better delayed impact. However, our method can still be applied to settings where the opposite is true—it suffices to change the sign of the delayed-impact observations associated with each instance.

---

> > ### Author Response · Authors · 2022-11-15
> > **Response/comments to reviewer 51vQ**
> >
> > **Claim that DI and DI fairness were not defined.**
> >
> > Thank you for your comment. We kindly emphasize, however, that all of these terms and concepts *were* formally defined.
> > - Delayed impact was first introduced, informally, in the Introduction, via an example: "*(...) the bank's decisions are informed by a classifier that predicts repayment success. These decisions may have a delayed impact in terms of the financial well-being of loan applicants, such as their savings rate or debt-to-income ratio, two years after a lending decision is made (...)  Suppose the bank deployed a classifier, which informed lending decisions, and logged the real-valued savings rate of each client two years later (i.e., logged the observed delayed impact associated with that client) (...) Importantly, ELF works with all measures of delayed impact that can be empirically observed or quantified*".
> > - In Section 2 (Problem Statement), we formally defined the random variable ($I$) associated with the observation of the delayed impact resulting from a particular prediction ($\widehat Y$) made by a classifier: "*Let $I^\beta_i$ be a real-valued delayed-impact observation resulting from deploying $\beta$ for the person described by the $i$-th data point. In our running example, $I^\beta_i$ corresponds to the empirically-observed savings rate two years after the prediction $\widehat Y^\beta_i$ was used to decide whether the $i$-th client should get a loan*".
> > - Finally, DI fairness was defined in Section 2 (Problem Statement): "*To define long-term [or delayed] fairness, we consider $k$ delayed-impact objectives, $g_j: \Theta \rightarrow \mathbb R$, $j \in \{1, \ldots, k\}$, that take as input the parameters, $\theta$, of a classifier, and return a real-valued measurement of fairness in terms of delayed impact. Then, we say that a classifier is fair in the long term iff $g_j(\theta) \leq 0$ for all $j$*".
> >
> > ---
> >
> > **Is Assumption 2 an assumption on the prediction-DI relationship?**
> >
> > Please notice that Assumption 2 does *not* correspond to an assumption about the relationship between predictions made by a model and their corresponding delayed impact, as suggested by the reviewer. Instead, it implies that the delayed impact of a decision depends *only* on the decision itself, not on the machine learning algorithm used to make it. For example, it makes no difference if an SVM or a neural network made a loan-repayment prediction; its DI depends only on whether the person actually received the loan.
> >
> > This means that our analyses hold no matter which underlying ML model is used to make predictions. This generality is key in real-life applications that may use different types of classifiers. A discussion on Assumption 2 (its intuitive meaning and implications)  is presented in Section 2 and further detailed in Appendix H.

---

> > > ### Comment · Reviewer_51vQ · 2022-12-03
> > > **Reply to the authors' response**
> > >
> > > I appreciate the authors' responses.
> > >
> > > 1. g and DI: It is usual to consider a function I > $\tau$ to be fair. 1)  existing work considers the difference between the two groups as unfair. The saving rate is not a good example. In most cases, the desired difference should be small, i.e. g>=0. It is unclear whether Hoeffding's inequality holds if the sign is changed.
> > >
> > > 2. DI is not well defined: my original question is about the relationship between I and DI. Most existing work considers the smaller difference between I implies better DI. But the definition of g and DI don't take the demographic groups into consideration. The difference between the two groups poses a new challenge: g has two I terms with difference signs: one for the favorable group while the other for the unfavorable group.
> > >
> > > 3. Thanks for the clarification about Assumption 2.

---

> > > > ### Author Response · Authors · 2022-12-14
> > > > **Response/comments to reviewer 51vQ**
> > > >
> > > > Thank you for your comments. As discussed below, our paper supports all alternative fairness definitions mentioned by the reviewer. Our method is also compatible with the mathematical formulations the reviewer argued are common in the literature.
> > > >
> > > > ---
> > > > ### Question 1:
> > > > - The reviewer is correct that it is usual to consider that I>$\tau$ implies safety. **That is precisely the formulation that we use in the paper (see line 2 of Section 3)**. This definition is compatible with our framework because:
> > > > 	- a) We defined (without loss of generality) that a solution $\theta$ is fair iff $g(\theta) \leq 0$.
> > > > 	- b) In Equation 1 of Section 3, $g(\theta)$ is defined as $\tau - E[I^{\pi_\theta} | C(X,Y,T)]$.
> > > > 	- c) Therefore, a fair solution would be one such that $\tau - E[I^{\pi_\theta} | C(X,Y,T)]\leq0$. This can be rewritten as $E[I^{\pi_\theta} | C(X,Y,T)] \geq \tau$, which is the form the reviewer argues is usual/common: that is, $I > \tau$.
> > > > - The reviewer points out that some of the existing work considers the difference between two demographic groups as being unfair. **Our framework supports this type of fairness definition, as discussed in Appendix E.3** ("*Enforcing general definitions of delayed-impact objectives*"). We further discuss this below.
> > > > - The reviewer argues that there are better examples than the savings rate, possibly because they believe we should focus on examples involving two demographic groups. Notice that this latter type of fairness is just **one** possible fairness definition studied in the literature—*and with which our method is fully compatible*. **In Appendix E.3 we discuss and provide an example of how our method can be used to ensure fairness with respect to two demographic groups**. Additionally, in Appendix A we provide other motivating examples for ELF.
> > > > 	- If the reviewer believes that some of these examples would be more compelling, we could move them to the main text.
> > > > - The reviewer says that in many practical applications, fairness could mean that the difference between statistics of two demographic groups is as small as possible. **Our framework supports this type of definition, as discussed in Appendix E.3**. In Appendix E.3, $\epsilon$ denotes how small the difference should be.
> > > > - The reviewer asks whether Hoeffding's inequality would hold if the sign of the delayed impact observation ($I$) is changed. Our method can be applied no matter what is the sign of $I$ (that is, independently of whether large values of $I$ correspond to better delayed impact or to worse delayed impact). In the latter case, it suffices to change the sign of the delayed-impact observation of each instance. **Hoeffding's inequality still holds in this case: bounds computed using Hoeffding's inequality do not assume or require that $I$ (or $E[I]$) is positive or negative**.
> > > > ---
> > > > ### Question 2:
> > > > - The reviewer argues that DI is not well defined. **Both $I$ and DI were defined in Section 2: $I$ is the random variable corresponding to the observation of the Delayed Impact (*DI*) resulting from a particular prediction made by the classifier**.
> > > > 	- In other words, Delayed Impact (**DI**) refers to the long-term effect we wish to consider when determining fairness (e.g., in Appendix A DI is the severity of chronic illnesses two years after a medical decision is made). The variable $I$, in this example, would represent one particular observation of the severity of the chronic illness of one particular patient two years in the future.
> > > > - The reviewer says that our definitions of $g$ and DI do not consider demographic groups. **This is not correct**: notice how the equation defining $g$ (Eq.1) is conditioned on $T$, which is the sensitive attribute representing, e.g., a demographic group.
> > > > 	- This is emphasized in our paper (Section 2) via a fairness definition example that explicitly takes into account the demographic group: $g(\theta) = \tau − E[I^{\pi_\theta} |T = A]$. This definition encodes the objective of ensuring fairness for people belonging to demographic group A. **Therefore, our definition of $g$ *does* take demographic group information into account**. If two demographic groups need to be considered, our method is *still* applicable—as discussed below and in the Appendix.
> > > > - The reviewer argues that fairness definitions considering the difference between two demographic groups pose a challenge to our method. **This is not correct**. In Appendix E.3 we present an example where fairness is defined with respect to two demographic groups. Here, the bank is interested in ensuring that the delayed impact of a classifier's predictions is approximately equal for loan applicants of demographic group A and group B. In this case, the DI objective is $g(\theta) =|E [I^{\pi_\theta}|T = A] − E [I^{\pi_\theta}|T = B]| - \epsilon$. This example (Appendix E.3), therefore, captures precisely the setting described by the reviewer: our method *is* naturally capable of dealing with this setting.

---

### Official Review · Reviewer_Sy63 · 2022-10-27

**Confidence:** 3
**Correctness:** 4
**Technical Novelty And Significance:** 2
**Empirical Novelty And Significance:** 2
**Recommendation:** 5

**Clarity, Quality, Novelty And Reproducibility:**

The writing of the paper is clear, the results are proved, and experiments are provided to back up the theory.

**Strength And Weaknesses:**

Strengths:
- the topic of the paper is important. I.e., how do we guarantee fairness across different individuals or groups, especially when we cannot observe labels about the individuals we make decisions on and update our model immediately?
- the approach is technically interesting: it is counterfactual in flavor and relies on learning from historical data. This data comes from previously deployed policies that may have been unfair, and the observed outcomes are informed by the previously deployed policy itself.
- the idea of moving away from having to model a specific analytical relationship between a classifier's prediction and its observed impact is nice

Weaknesses:

I have the following concerns about the paper:

i) Exposition and scope-wise, I am not sure whether there is anything in the paper that is specific to delayed impact and but not to counterfactual learning. Here, unless I missed something about the paper, it seems that the only way the delayed impact really comes in is that compared to a standard ML/fairness model, the observed label is defined as being, say, 2 years after the decision has been made. But the time dimension here does not actually play a role in the model:
- the dataset itself is static, coming from a single previously deployed policy.
- The problem is, in fact, formulated as a static problem of counterfactual learning given historical i.i.d. data. There isn't a real time dimension/aspect in which the model is updated over time and more data is collected over time.

In turn, it seems to be more of a traditional counterfactual learning problem where the delayed impact part feels "gimmicky" and does not impact the paper beyond its motivation.

ii) Assumption 1 seems to essentially say that we look at models whose predictions do not depend on group identity. This is a "sensitive attribute-blind" approach which contradicts much of what is known in the space of fairness. I.e., the same features may mean different things for different populations, and not taking the sensitive attribute into account is a surefire way to promote unfairness. This seems like a bit of a naive assumption which says that we can take the lessons from one population and apply it to another directly.

iii) I think the assumptions that the historical data is well-behaved (i.i.d and having observations of the full support) may be a bit hard to believe. What I think may be hard and interesting about time-dynamics in the space of fairness is that the deployed rule influences the observed data, and the observed data influences the deployed rule, often leading to feedback loop with a long-term negative impact.

As an example: if we only historically gave loans to wealthy white people, a naive model will learn to still only give loans to wealthy white people, and we will only collect repayment data from wealthy white people. In this case, we do not have access to i.i.d., full support data that is representative of the entire population, and have a missing data problem. It is then impossible (without collecting more data) to predict the outcomes on a different population that we have observed (almost) nothing about in the past.




**Summary Of The Paper:**

The paper looks at how to guarantee fairness when the impact of the decisions we make today is only seen in the long term. The paper provides a counter-factual approach to predict future impact from historical data that may be coming from a different policy than the one we aim to deploy, and aims to find a policy that does not have an unfair long-term impact across populations.

**Summary Of The Review:**

I think the topic of the paper is interesting. However, I think I feel that much of what makes delayed-impact fairness hard in practice is in the details and specific assumptions made about the problem (non i.i.d. data and data from a single population that cannot be interpolated to a different population), and I think the assumptions of the current work are a bit naive. As such, I think the paper is a bit under the bar in its current form

---

> ### Author Response · Authors · 2022-11-15
> **Response/comments to reviewer Sy63**
>
> We thank the reviewer for their positive comments on ELF's novelty/importance: unlike existing methods, ELF does not require knowledge of the analytical relationship between a classifier's predictions and its observed delayed impact. We appreciate their remarks on the interesting use of counterfactual techniques to propose the first method capable of ensuring delayed-impact fairness guarantees with high confidence, based only on historical data. We are encouraged by their comments on the relevance of our problem formulation to real-life settings, where only data from past classifiers is available.
>
> We appreciate the reviewer's constructive comments/suggestions. We address their questions below.
>
> ---
>
> **On the connection between ELF and counterfactual learning.**
>
> We thank the reviewer for bringing up the connection between our method and counterfactual learning. The reviewer is correct that our paper explores this connection. One of the key insights of the paper is that enforcing delayed impact fairness without analytical models can be posed as a counterfactual learning (or off-policy evaluation) problem. In particular, we use counterfactual reasoning (in the form of importance sampling) to design a novel supervised learning algorithm that solves an important open problem—ensuring delayed impact fairness based only on historical data.
>
> ELF uses counterfactual reasoning to estimate the delayed impact had we deployed a particular novel classifier. It is the first algorithm capable of doing so without having to deploy the new classifier to verify whether it is fair and without requiring an analytic model of the delayed impact caused by a classifier's predictions.
>
> The reviewer is correct that our method may be adapted to provide high probability guarantees on *other* metrics, instead of only metrics related to delayed impact fairness. We will update the paper to emphasize that our method *is* a type of counterfactual learning mechanism—one designed specifically to tackle a particular open problem in the long-term fairness literature.
>
> ---
>
> **How does ELF extend the standard supervised learning setting?**
>
> We would like to clarify that in our setting, labels $Y$ are defined exactly as they are in standard supervised learning. Our setting differs from standard supervised learning in that predictions, $\widehat Y$, made by a classifier, may have a delayed impact, $I$. Here, $I$ is any empirically observable real-value quantity that may be affected by predictions—such as the savings rate of a client two years after a lending decision (informed by predictions $\widehat Y$ of repayment success) is made.
>
> Notice that delayed impact, $I$, is different from the label, $Y$, of an instance. While the label is always available and used (as in standard supervised learning) to train a classifier, the delayed impact $I$ can only be observed after deploying a given model. Therefore, our setting *extends* standard supervised learning: it still considers labels ($Y$) and label predictions ($\widehat Y$), but also another key quantity—the delayed impact $I$ resulting from predictions $\widehat Y$.
>
> ---
>
> **Delayed-impact fairness under sequences of interdependent classifiers.**
>
> The reviewer mentioned the *sequential fair decision-making setting*, where sequences of classifiers are deployed and updated over time. This setting, where long-term fairness implies ensuring that *interdependent* classifiers are fair, is an important future research direction.
>
> To the best of our knowledge, this is an open problem if a model is not known. The first step to solving the model-free multi-step fairness problem is addressing the corresponding one-step setting. ELF is the *first* principled method capable of doing so, and is, therefore, an important first step toward achieving the goal mentioned by the reviewer. Extending ELF to these different settings is an important and interesting direction for future work. Our paper discusses this important related problem in the Introduction, Related Work, and Appendix K. In summary, existing methods in the relevant literature either:
> - ensure that *static* fairness constraints hold over time (e.g., Wen et al., 2019). Importantly, computing static constraints require analytic expressions based on a model's predictions.
>  - ensure long-term fairness in *sequential* decision problems where the system's model is known (e.g., Zhang et al., 2020a).
>
> Our main algorithmic contribution, by contrast, addresses a *different* problem. We study the orthogonal setting where models are not known and long-term fairness constraints cannot be computed analytically. ELF is the first method that works under the less strict assumption that only historical data is available; i.e., it only has access to observations of the delayed impact resulting from an existing classifier's predictions.

---

> > ### Author Response · Authors · 2022-11-15
> > **Response/comments to reviewer Sy63**
> >
> > **Using data from multiple previously-deployed models.**
> >
> > Thank you for bringing this point up. ELF can be readily adapted if data comes from multiple models. When computing importance sampling estimators (line 5, Alg.1), it suffices to replace $\beta$ with the specific classifier used to make the prediction $\widehat Y$ of each corresponding instance.
> >
> > ---
> >
> > **Sensitive attribute-blind models.**
> >
> > The reviewer is correct that some fairness definitions in the literature are not "sensitive attribute-blind". Importantly, our method can handle such definitions of fairness as well. In this case, it would suffice to include $T$ (the sensitive attribute) as another attribute of $X$ (the set of attributes used to make predictions). The assumption that only $X$ is given as input to the classifier was made only for notational clarity in the subsequent proofs. It does *not* constrain our method to the "sensitive attribute-blind" setting. In particular, the practical implication of Assumption 1 is that the classifier does not *have* to depend on $T$ (which would be the case if tackling the "sensitive attribute-blind" setting). The model *can*, however, depend on such attribute—it suffices to add it to $X$. Assumption 1, thus, is consistent with our method being applied in both settings (blind and not blind).
> >
> > ---
> >
> > **Assumption of support and i.i.d. data.**
> >
> > The reviewer mentioned that a weakness of our method is the assumption that data instances are i.i.d. We would like to point out that this assumption is standard in the supervised learning literature (Bishop, 2006). It is not an assumption particular to our method—essentially all supervised learning methods make the same assumption.
> >
> > Additionally, note that full support is a property of the classifier, not the dataset itself. This assumption is common in the reinforcement learning community when designing counterfactual methods based on importance sampling. In our setting, this assumption can be trivially satisfied when ELF uses standard stochastic classifiers that place non-zero probability on its outputs; e.g., when using any modern neural network classifiers with a Softmax layer, which are commonplace in the literature.
> >
> > ---
> >
> > **Settings where the delayed impact of some decisions may not be available.**
> >
> > The one-sided setting described by the reviewer is important, and we discuss it in the Introduction of our paper. We would like to point out that:
> > - Our method *can* be applied to a wide range of problems where representative historical datasets can be constructed. In addition to the examples used in our empirical evaluation (which uses real-life data), we discuss, in Appendix A, other real-life motivating examples of ELF. For instance: *(1)* Consider a university deciding which students should receive 1-on-1 tutoring, based on GPA predictions. DI could be the likelihood that students with different backgrounds will graduate from college; *(2)* Consider a police department deciding which crime prevention strategy to use, based on predictions about crime recidivism. DI could be the average incarceration rate two years after this decision; it could indicate, e.g., that people of a particular race are unfairly affected. See Appendix A for a detailed discussion and additional examples.
> > - Importantly, the assumption of a representative dataset is *significantly* less restrictive than the assumption that the algorithm has access to an accurate analytical model of the prediction-DI relationship. In particular, notice that in the one-sided setting described by the reviewer, existing techniques would require access to a model describing the delayed impact of any predictions *both* on wealthy white people *and* on people of other races/social classes. Even though constructing such models is challenging, one could argue that they may be learned based on data. Notice, however, that this would *still* require access to data from clients of all races and social classes—as the reviewer correctly pointed out. In other words, requiring access to training data on the delayed impact affecting different types of people is, thus, unavoidable. Since estimating models is challenging, our method was specifically designed to provide high-confidence fairness guarantees even when analytical models are not known.
> >
> > We agree that tackling the one-sided dataset setting is an important future direction, but it is outside the scope of this paper.

---

### Author Response · Authors · 2022-11-15
**General Comments to All Reviewers**

We thank the reviewers for their thorough feedback and suggestions. In particular:

- We are encouraged by the reviewers' positive comments on the novelty and significance of our method (ELF). ELF is the first classification algorithm that provides high-confidence fairness guarantees in terms of long-term, or delayed, impact, when an analytical model describing the relationship between predictions and delayed impact is not known.
- We appreciate their remarks on the interesting use of counterfactual techniques to propose the first method capable of ensuring delayed-impact fairness guarantees with high confidence, based only on historical data. This challenge was, up to this point, an open problem.
- We are encouraged by their comments on the relevance of our problem formulation to real-life settings, where only data from past classifiers is available.
- We thank reviewers for their positive comments on the novelty of our approach, which simultaneously formulates the fair classification problem as both a classification and a reinforcement learning problem. Furthermore, we appreciate their comments on the formal guarantees of our algorithm—it is proven to identify nontrivial solutions (fair classifiers) with high confidence.

The reviewer's concerns can all be classified into requests for clarifications, potential extensions to our work, and grounding of assumptions. Below, we:
- Address the questions brought up by reviewers and their requests for clarification—all of which can be easily added to the paper before publication.
- Discuss ways in which our method may be extended to tackle the alternative settings mentioned by reviewers.
- Further discuss why each of our assumptions is common in related work and holds in reasonable settings, including in the multiple real-world scenarios described in Appendix A.

---

### Decision · Program_Chairs · 2023-01-20

**Decision:**

Reject

**Justification For Why Not Higher Score:**

Unanimously slightly below the bar, no champion.

**Justification For Why Not Lower Score:**

N/A

**Metareview: Summary, Strengths And Weaknesses:**

The delayed impact of fairness is an important and understudied problem.
However, the assumptions, empirical evaluation, and some technical details have been brought into question. No reviewer was championing the paper.
The author response is very detailed and thorough. However, given the importance of this type of work, the meta-reviewer believes it is necessary to incorporate these various clarifications into a revision of the paper for another round of reviewing.

---

> ### Author Response · Authors · 2023-02-03
> **Response to Paper Decision**
>
> Thank you for the hard work in evaluating the large number of submissions to
> this conference. However, for completeness, we feel that we must state that
> we strongly disagree with this summary of our work.
>
> This is an unfortunate case of reviewers doing relatively shallow reviews
> (undoubtedly because of a heavy reviewing load) and making factually false
> claims. In their rebuttal, the authors pointed out the numerous factual (and
> easily refutable) errors, but the reviewers did not respond and failed to
> engage in any discussion with the authors.
>
> These errors included:
> * claims that definitions were missing from the manuscript when the definitions were right there (the rebuttal included specific line numbers);
> * claims that assumptions are unusual when nearly all related papers in the field make the same assumptions;
> * inexplicable concerns that our work can solve other related problems, in addition to the one we solve; and
> * concerns that we do not compare to existing model-free methods, when we present the first and only model-free method to date, and the reviewer points out no actual counterexample technique to which we should or even could compare.
>
> Unfortunately, the reviewers ignored the authors' responses to their concerns
> and factually false claims that were easily refutable. In the end, we feel
> strongly that the paper was rejected for flawed reasons, and that the
> reviewers simply did not engage with the reasoned arguments pointing out the
> paper's value.